# Expert Merging: Model Merging with Unsupervised Expert Alignment and Importance-Guided Layer Chunking

**Dengming Zhang**[1,2*]  **Xiaowen Ma**[1*]  **Zhenliang Ni**[1]
**Zhenkai Wu**[1,2]  **Han Shu**[1]  **Xin Jiang**[1]  **Xinghao Chen**[1†]
[1] Huawei Technologies    [2] Zhejiang University

## Abstract

Model merging, which combines multiple domain-specialized experts into a single model, offers a practical path to endow Large Language Models (LLMs) and Multimodal Large Language Models (MLLMs) with broad capabilities without the cost of joint training or serving many models. However, training-free methods rely on hand-tuned coefficients, whereas training-based methods primarily align parameters rather than downstream task behavior and typically treat all layers uniformly, ignoring inter-layer heterogeneity. We introduce Expert Merging, a training-light method that learns a small set of layer-wise coefficients using only unlabeled calibration data. The coefficients are optimized to explicitly align the merged model's hidden states and logits with those of the corresponding experts, with a coefficient regularizer for stability and task-weighted losses for controllable trade-offs. To capture inter-layer variation, Expert Merging++ augments this design with importance-guided chunking: a normalized layer-importance metric, derived from learned coefficients, task-vector magnitudes, and parameter counts, allocates more chunk-wise coefficients to high-importance layers while keeping low-importance layers lightweight. The result is a label-free, parameter-efficient, and scalable approach to multi-expert model merging across LLMs and MLLMs. Across MLLM backbones (InternVL and Qwen2-VL) and the LLM backbone (Mistral), our method surpasses strong training-free and training-based merging baselines, with Expert Merging++ delivering further gains and, in some cases, even exceeding supervised Mixture Training. Our code is available at `https://github.com/Littleor/ExpertMerging`.

## 1 Introduction

Large language models (LLMs) show strong general abilities but often lag on domain-specific tasks that require specialized knowledge (Hadi et al., 2023; Ling et al., 2023). A standard remedy is Supervised Fine-Tuning (SFT) that turns a base model into domain-specific expert models for code, math, or reasoning (Zhao et al., 2023; Dodge et al., 2020; Chung et al., 2024). However, maintaining and serving a separate model per domain imposes substantial storage, memory, and engineering overhead, hindering deployment at scale (Yadav et al., 2024). Model merging offers an attractive alternative: combine multiple domain experts into a single multi-competent model while retaining their respective strengths (Yu et al., 2024; Wortsman et al., 2022; Ilharco et al., 2022). Existing model merging methods broadly fall into two groups: training-free methods with predefined coefficients and training-based methods that learn coefficients or parameters (Wei et al., 2025).

Among training-free approaches, Task Arithmetic (Ilharco et al., 2022) as a classic method forms a merged model by adding expert task vectors (i.e., subtracting the base model parameters from the SFT model parameters) to the base model with fixed, predefined coefficients. Building on this idea, subsequent training-free approaches (Yadav et al., 2023; Yu et al., 2024; Gargiulo et al., 2025) refine the merging process to mitigate parameter redundancy and interference, reporting measurable

---

[*]Equal contribution.
[†]Corresponding author: xinghao.chen@huawei.com

improvements. However, these methods typically rely on hand-specified hyperparameters (e.g., per-task coefficients), which are selected via manual tuning or grid search. More importantly, these methods often focus solely on parameter-space alignment while neglecting task alignment.

Training-based methods address these limitations by learning merging parameters or coefficients through gradient-based optimization. WUDI (Cheng et al., 2025) reduces cross-task interference by minimizing a layer-wise interference objective, improving merged model performance. Building on this idea, WUDI v2 (Wei et al., 2025) performs singular value decomposition (SVD) before training to isolate task-salient signals from noise, further enhancing performance. Nevertheless, these approaches remain primarily parameter-space alignment procedures and overlook the merged model's downstream performance. AdaMerging (Yang et al., 2024) optimizes task-wise or layer-wise coefficients by minimizing prediction entropy to obtain a favorable combination of coefficients. While this accounts for task performance, entropy minimization provides no explicit task alignment signal and can induce overconfidence under distribution shift; moreover, assigning a single trainable coefficient per layer ignores inter-layer heterogeneity, ultimately constraining performance.

To address both task alignment and inter-layer heterogeneity, we introduce Expert Merging and its extension, Expert Merging++. For task alignment, Expert Merging learns trainable layer-wise coefficients that align the merged model's hidden states and logits with those of the corresponding expert models on a small set of unlabeled inputs (5–10 samples), thereby explicitly matching downstream behavior. To mitigate distribution shifts during merging, we incorporate a coefficient regularization term that stabilizes optimization. For inter-layer heterogeneity, our analysis reveals substantial variation across layers in optimal coefficient magnitudes and parameter number. We therefore propose a layer-importance metric to quantify each layer's contribution and a chunk-wise coefficient method: high-importance layers are partitioned into structured parameter chunks with distinct coefficients, while low-importance layers remain lightweight. We evaluate these methods on LLMs and multimodal large language models (MLLMs): Expert Merging consistently outperforms strong training-free and prior training-based baselines in both settings, Expert Merging++ yields additional gains, and on MLLMs the approach can even surpass supervised Mixture Training.

Our contributions are: (1) Expert Merging that learns layer-wise coefficients to align the merged model's hidden states and logits to expert models on unlabeled data, with coefficient regularization for stability; (2) Expert Merging++ with chunk-wise coefficients guided by a layer-importance metric to allocate more capacity by chunking high-importance layers while keeping low-importance layers lightweight; and (3) comprehensive results and ablations on both an LLM (Mistral-7B) and MLLMs (InternVL, Qwen2-VL) showing Expert Merging surpasses training-free and prior training-based baselines, with Expert Merging++ yielding further gains.

## 2 RELATED WORK

Model merging aims to combine multiple task-specific models into a unified model with diverse capabilities (Wortsman et al., 2022). This approach has gained significant attention in the large model domain, particularly for large language models (Wei et al., 2025; Zhou et al., 2024; Goddard et al., 2024), due to not requiring access to original training datasets and its cost-effective nature (Stoica et al., 2023). Existing model merging approaches can be categorized into two main classes based on whether they require training: training-free methods and training-based methods (Wei et al., 2025).

### 2.1 TRAINING-FREE MODEL MERGING METHODS

Training-free methods (Wortsman et al., 2022; Gargiulo et al., 2025; Marczak et al., 2025) operate directly on model parameters without requiring additional optimization procedures. These approaches typically manipulate pre-trained model weights through mathematical operations to achieve effective model combination. Task Arithmetic (Ilharco et al., 2022) represents a foundational approach in this category, computing task vectors for each task-specific model by subtracting the base model parameters from fine-tuned parameters. The method then merges multiple task-specific models through predefined scaling coefficients. However, this approach often suffers from parameter interference when combining multiple tasks. TIES Merging (Yadav et al., 2023) addresses the parameter conflict problem inherent in Task Arithmetic by proposing a sophisticated conflict resolution mechanism. The method analyzes parameter changes based on both magnitude and di-

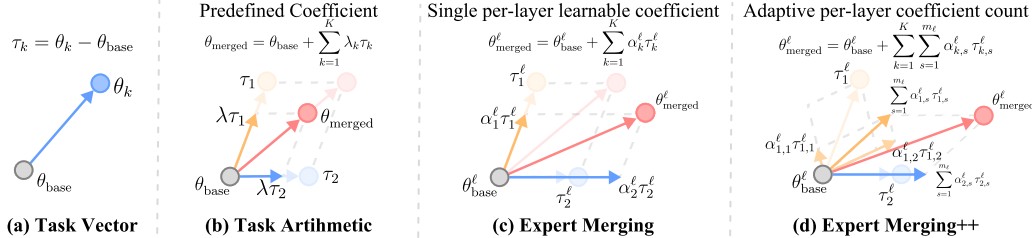

Figure 1: Conceptual overview: (a) Task Vector, (b) Task Arithmetic (TA), (c) Expert Merging, and (d) Expert Merging++ with importance-guided chunk-wise coefficients.

rection, trimming conflicting parameters before merging. Specifically, TIES identifies parameters with conflicting signs across different task vectors and resolves these conflicts through magnitude-based voting schemes. DARE (Yu et al., 2024) introduces a different perspective by focusing on parameter sparsification during the merging process. The method applies drop and rescale operations to task vectors, randomly dropping a fraction of parameters and rescaling the remaining ones to maintain the expected magnitude. This approach reduces parameter interference by randomly dropping parameters, demonstrating that sparsification can actually improve merging performance. Despite their simplicity, training-free approaches typically rely on fixed, hand-tuned coefficients and operate purely in parameter space without explicit task-level alignment, making them sensitive to hyperparameters and prone to degraded performance when many experts are combined.

## 2.2 Training-Based Model Merging Methods

Training-based methods optimize model combination through gradient-based updates, enabling more adaptive merging strategies. These methods can be further categorized by their data requirements. Sample-independent methods eliminate the dependency on training samples while still employing optimization. WUDI (Cheng et al., 2025) reduces cross-task interference by minimizing a layer-wise interference objective, aligning the merged model's activations with those of the experts and improving robustness. Building on this idea, WUDI v2 (Wei et al., 2025) performs a singular value decomposition (SVD) of task vectors before training to isolate task-salient signals from noise, further improving performance. Sample-dependent methods leverage data to optimize merging parameters. AdaMerging (Yang et al., 2024) optimizes task-wise or layer-wise coefficients by minimizing prediction entropy to obtain a favorable combination of coefficients. While effective, entropy minimization provides no explicit task-alignment signal and can induce overconfidence under distribution shifts; moreover, assigning a single trainable coefficient per layer overlooks inter-layer heterogeneity, which can limit performance.

Despite progress, most training-based methods emphasize parameter-space alignment or implicit distribution matching, provide limited control over task trade-offs, and ignore layer heterogeneity. We instead align hidden states/logits on unlabeled data with regularized layer-wise and importance-guided chunk-wise coefficients, improving alignment, trade-off control, and performance.

## 3 Method

We propose Expert Merging, a training-light method that merges multiple domain experts via layer-wise coefficients learned from unlabeled data to align hidden states and logits; Expert Merging++ further allocates capacity by layer importance with chunk-wise coefficients (Figure 1).

### 3.1 Preliminaries

Let $\theta_{\text{base}}$ denote the parameters of a base model $f(\cdot; \theta_{\text{base}})$ with $L$ layers, and let $\{\theta_k\}_{k=1}^K$ be the parameters of $K$ domain experts obtained by SFT on tasks $\{\mathcal{T}_k\}_{k=1}^K$. Define the task vector of expert $k$ as $\tau_k = \theta_k - \theta_{\text{base}}$ and its layer-$\ell$ block as $\tau_k^\ell$. Training-free Task Arithmetic forms a merged model by $\theta_{\text{TA}} = \theta_{\text{base}} + \sum_{k=1}^K \lambda_k \tau_k$ with predefined coefficients $\{\lambda_k\}$ (panel (b) of Figure 1). In contrast, we seek *trainable, layer-wise/chunk-wise* coefficients that are learned from unlabeled data.

We assume access to an unlabeled, task-partitioned calibration set $\mathcal{D} = \bigcup_{k=1}^{K} \mathcal{D}_k$ where $\mathcal{D}_k$ contains inputs reflecting the distribution of task $\mathcal{T}_k$ (text-only for LLMs; image+text for MLLMs). From $\mathcal{D}$ we learn *layer-wise* (and later *chunk-wise*) coefficients, with the objective of aligning the merged model's performance with expert $k$ on inputs $x \in \mathcal{D}_k$.

**Limitations of existing trainable merging.** Task vector based methods such as AdaMerging and WUDI Merging also introduce trainable coefficients on top of Task Arithmetic, but their objectives do not explicitly align the merged model with each expert on its own domain. For example, AdaMerging writes the merged parameters as $\theta_{\text{Ada}}(\lambda) = \theta_{\text{base}} + \sum_{k=1}^{K} \lambda_k \tau_k$, and optimizes the coefficients on multi task unlabeled data via entropy minimization

$$\mathcal{L}_{\text{Ada}}(\lambda) = \sum_{k=1}^{K} \sum_{x \in \mathcal{B}_k} H\big(p(y \mid x; \theta_{\text{Ada}}(\lambda))\big), \tag{1}$$

where $H(\cdot)$ is Shannon entropy and $\mathcal{B}_k \subset \mathcal{D}_k$ is a batch of unlabeled inputs from task $k$. Minimizing $\mathcal{L}_{\text{Ada}}$ mainly pushes the merged model to make high confidence predictions, because any very peaked distribution has low entropy, even if the model becomes very confident about an answer that expert $k$ would not choose. In particular, very different logit distributions can share the same entropy, so the model can decrease $\mathcal{L}_{\text{Ada}}$ by becoming confidently wrong. Moreover, the entropy is averaged over all tasks, so the coefficients can be driven by tasks where confidence is easiest to increase, without any term that requires $p(y \mid x; \theta_{\text{Ada}}(\lambda))$ to align with the expert distribution on $\mathcal{D}_k$. As a result, some experts may be underrepresented even when they are strong on their own domains.

WUDI Merging instead analyzes the interference between the merged model and each expert at selected linear layers. Let $\theta_k^{\ell}$ and $\theta_{\text{merged}}^{\ell}$ denote the parameters of layer $\ell$ for expert $k$ and the merged model, and let $x_{k,\ell}$ be the corresponding layer input. The interference on task $k$ at layer $\ell$ can be written as

$$\mathcal{J}_{k,\ell} = \mathbb{E}_{x_{k,\ell}} \big\| \theta_{\text{merged}}^{\ell} x_{k,\ell} - \theta_k^{\ell} x_{k,\ell} \big\|_2^2, \tag{2}$$

and WUDI Merging seeks a merged model that minimizes this interference aggregated over tasks and layers. This controls a local linear discrepancy, but it is defined only on selected linear layers and does not directly match the full hidden trajectories or the predictive distributions, so models with small interference can still produce noticeably different predictions on complex, multi-modal tasks. These limitations motivate us to directly align the merged model with each expert at both the representation and prediction levels on task-partitioned calibration data, which leads to the hidden state and logit alignment losses in equation 3 and equation 4.

### 3.2 EXPERT MERGING

The key idea of Expert Merging is to learn per-layer coefficients by explicitly aligning the merged model's internal representations and predictions to those of the experts on unlabeled inputs from their respective domains. The end-to-end training pipeline is depicted in Figure 2: the base and expert models are frozen, the coefficients are the only trainable parameters, and alignment is applied on both hidden states and logits.

**Parameterization.** Expert Merging uses one trainable coefficient per layer (panel (c) of Figure 1). For each layer $\ell$, we set $\theta_{\text{merged}}^{\ell} = \theta_{\text{base}}^{\ell} + \sum_{k=1}^{K} \alpha_k^{\ell} \tau_k^{\ell}$, where $\alpha_k^{\ell} \in \mathbb{R}$ is a learnable coefficient for expert $k$ at layer $\ell$. This yields only $K \times L$ trainable coefficients, initialized with a fixed value $\bar{\alpha}_k$.

**Alignment losses.** For an input $x$, we denote the merged model's hidden state at layer $\ell$ by $\mathbf{h}_{\ell}(x; \theta_{\text{merged}})$ and its pre-softmax logits by $\mathbf{z}(x; \theta_{\text{merged}})$. Likewise, $\mathbf{h}_{\ell}^{(k)}(x)$ and $\mathbf{z}^{(k)}(x)$ denote the corresponding values from expert $k$. Specifically, for inputs $x \in \mathcal{D}_k$ of task $k$, we minimize a combination of hidden-state and logit alignment (green and yellow dashed connections in Figure 2):

$$\mathcal{L}_{\text{hid}}^{(k)} = \sum_{\ell \in \mathcal{S}} \mathbb{E}_{x \sim \mathcal{D}_k} \big\| \mathbf{h}_{\ell}(x; \theta_{\text{merged}}) - \mathbf{h}_{\ell}^{(k)}(x) \big\|_2^2, \tag{3}$$

$$\mathcal{L}_{\text{logit}}^{(k)} = T^2 \, \mathbb{E}_{x \sim \mathcal{D}_k} \, \text{KL}\Big( \text{softmax}(\mathbf{z}^{(k)}(x)/T) \, \big\| \, \text{softmax}(\mathbf{z}(x; \theta_{\text{merged}})/T) \Big), \tag{4}$$

where $\mathcal{S}$ is a subset of layers used for hidden alignment (e.g., every Transformer layer's output), and $T$ is the temperature. Hidden-state matching enforces intermediate alignment, and logit alignment

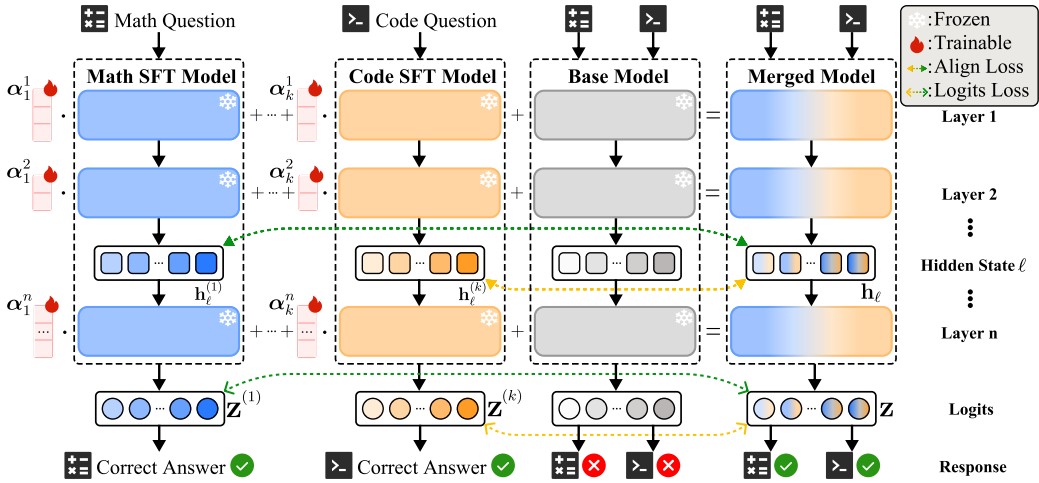

Figure 2: Architecture of Expert Merging++. The base and expert models are frozen (snowflakes) and only the chunk-wise coefficients $\{\boldsymbol{\alpha}_k^\ell\}$ are trainable (flames). For unlabeled inputs from each domain, the merged model aligns both hidden states $\mathbf{h}_\ell$ (green) and logits $\mathbf{z}$ (yellow) to the corresponding expert, producing a single model that preserves all task performance.

promotes task-faithful outputs, which are crucial for downstream performance. The logit loss is a multi task alignment objective that minimizes the Kullback-Leibler divergence between the expert distribution and the merged model distribution on $\mathcal{D}_k$, so when experts are accurate this directly reduces the supervised loss without requiring labels. In contrast to entropy based objectives that only push predictions to be confident, equation 4 encourages the merged model to align with the expert on which predictions should be confident. The hidden state loss complements this by matching the full representation trajectories $\{\mathbf{h}_\ell(x)\}_{\ell \in \mathcal{S}}$, which implicitly controls layer level interference terms such as $\|\theta_{\text{merged}}^\ell x_{k,\ell} - \theta_k^\ell x_{k,\ell}\|_2^2$ while also constraining non linear layers that are not covered by the original WUDI analysis.

**Controllable task trade-offs.** Unlike prior trainable merging that implicitly balances tasks without offering any external control, we introduce explicit nonnegative task weights $\{\beta_k\}$ to control domain priority. Conceptually, each task contributes a weighted alignment signal: $\mathcal{L}_{\text{align}} = \sum_{k=1}^K \beta_k \left( \mathcal{L}_{\text{hid}}^{(k)} + \mathcal{L}_{\text{logit}}^{(k)} \right)$ with $\beta_k \geq 0$. Setting $\beta_k$ modulates how strongly expert $k$ is preserved, providing direct, interpretable control over cross-task trade-offs. This explicit weighting makes the trade-offs transparent and tunable, addressing the concern that entropy based or interference based objectives may implicitly favor some tasks over others without exposing a clear control knob.

**Coefficient regularization.** To stabilize optimization and mitigate distribution shifts, we regularize the coefficients around the initial value $\bar{\alpha}_k$:

$$\mathcal{R}(\alpha) = \frac{1}{KL} \sum_{k=1}^K \sum_{\ell=1}^L \left| \alpha_k^\ell - \bar{\alpha}_k \right|. \tag{5}$$

The overall objective is $\min_{\{\alpha_k^\ell\}} \ \mathcal{L}_{\text{align}} + \gamma \mathcal{R}(\alpha)$, where $\gamma$ controls regularization strength. Furthermore, since the initial values have a significant impact on the optimization process, we use coefficients validated by Task Arithmetic as the initial values. This regularizer can be viewed as a prior that keeps the solution close to the training free Task Arithmetic point, and only allows coefficients to move away when the hidden and logit alignment losses provide consistent evidence. In practice this prevents overfitting the small calibration sets and avoids degenerate solutions that might arise if we optimized the coefficients using only entropy based or interference based objectives.

### 3.3 EXPERT MERGING++

Layer-wise coefficients assume uniform importance across layers and parameter groups, which is violated in practice: (i) different layer types (e.g., attention vs. MLP) have very different parameter

counts; (ii) even within a type, deeper layers can be more influential; and (iii) tasks impact layers unevenly. Expert Merging++ addresses this by estimating layer importance and allocating finer-grained, chunk-wise coefficients where they matter most (panel (d) of Figure 1).

**Layer-importance metric.** After training Expert Merging, we compute a normalized importance score per layer based on three factors: learned coefficient magnitude, task-vector weight, and parameter count. Specifically, let $s_k^\ell$ denote the task-vector weight of expert $k$ at layer $\ell$, computed from the SFT task vector (e.g., mean absolute magnitude $s_k^\ell = \text{mean}(|\tau_k^\ell|)$) and normalized across layers. Then $I_\ell = \text{Norm}\left(\sum_{k=1}^K |\alpha_k^\ell| s_k^\ell n_\ell\right)$, where $n_\ell = \text{param\_count}(\ell)$ is the number of parameters in layer $\ell$, and $\text{Norm}(\cdot)$ denotes $\ell_1$ normalization to $[0, 1]$ across layers. Intuitively, $I_\ell$ is a data driven proxy for how much the experts and the learned coefficients rely on layer $\ell$ across tasks, so it gives a principled signal for where additional merging capacity is likely to yield the largest gains.

**Importance-guided chunk-wise coefficients.** As we have calculated the importance of different layers $I_\ell$, the next step is to chunk the layers according to their importance, with more chunks representing more trainable coefficients. Given a total coefficient budget $B$ for each task, we allocate to layer $\ell$ a chunk count

$$m_\ell = \left\lfloor B \frac{I_\ell^\kappa}{\sum_{j=1}^L I_j^\kappa} \right\rfloor, \qquad \kappa \geq 0, \tag{6}$$

and partition its parameter tensors into $m_\ell$ parameter chunks. Here $\kappa$ is an allocation steepness hyperparameter: $\kappa = 0$ yields near-uniform allocation; $\kappa = 1$ allocates proportional to importance; $\kappa > 1$ concentrates chunks on high-importance layers; $0 < \kappa < 1$ flattens the distribution. In implementation, we flatten the tensor and split it into $m_\ell$ contiguous chunks. Each chunk $s$ receives a distinct trainable coefficient $\alpha_{k,s}^\ell$ for expert $k$. For some low-importance layers, we set $m_\ell = 0$ and use a fixed, untrainable scalar, preserving a compact parameterization.

**Objective.** We reuse the alignment objective with chunk-wise coefficients as in equation 3 and equation 4. Let $\mathcal{S}_{\ell,s}(\cdot)$ denote a chunk-selection operator that extracts chunk $s$ from the layer $\ell$, and define $\tau_{k,s}^\ell = \mathcal{S}_{\ell,s}(\tau_k^\ell)$. For each chunk $\tau_{k,s}^\ell$, we will have a trainable chunk-wise coefficient $\alpha_{k,s}^\ell$. Then

$$\theta_{\text{merged}}^\ell = \theta_{\text{base}}^\ell + \sum_{k=1}^K \sum_{s=1}^{m_\ell} \alpha_{k,s}^\ell \tau_{k,s}^\ell. \tag{7}$$

For brevity set $\boldsymbol{\alpha}_k^\ell = (\alpha_{k,1}^\ell, \ldots, \alpha_{k,m_\ell}^\ell)$ and $\boldsymbol{\tau}_k^\ell = (\tau_{k,1}^\ell, \ldots, \tau_{k,m_\ell}^\ell)$, a shorthand also used in Figure 2. The loss is identical to Section 3.2, replacing $\alpha_k^\ell$ by $\alpha_{k,s}^\ell$. Regularization includes a chunk-aware term $\frac{1}{BK} \sum_{k,\ell,s} |\alpha_{k,s}^\ell - \bar{\alpha}_k|$ which prevents the chunk-wise coefficients from deviating excessively from their initial distribution.

Importance-guided chunking allocates learnable capacity to layers that matter most, improving performance especially when there are significant differences among experts or layers (as observed in our experiments). Moreover, since the value of $B$ remains close to 1 in practice (0.9-1.2), even the number of chunk-wise coefficients is nearly the same as that of the layer-wise coefficients, thereby preserving strong sparsity in the merged model while improving the performance. Notably, Expert Merging++ is an enhanced version of Expert Merging. In practice, we first run Expert Merging to learn layer-wise coefficients, then use these learned coefficients to compute the layer-importance scores $I_\ell$ and determine how many chunks $m_\ell$ each layer receives. In the second stage, we keep the base and expert models frozen and optimize only the chunk-wise coefficients with the same hidden-state and logit alignment objectives as before.

# 4 EXPERIMENT

## 4.1 SETUP

**Backbones.** We evaluate our method on both LLMs and MLLMs. For LLMs, we use Mistral-7B-v0.1 (Jiang et al., 2023) as the base model and use experts specialized for three capabilities: instruc-

tion following (Chat)[1], mathematical reasoning (Math)[2], and code generation (Code)[3]. For MLLMs, following prior merging methods (Wei et al., 2025), we adopt InternVL2.5-1B-Instruct (Chen et al., 2024) and Qwen2-VL-7B-Base (Wang et al., 2024b) as backbones, and use experts for five capabilities: visual question answering (VQA), geometry reasoning (Geometry), chart understanding (Chart), optical character recognition (OCR), and visual grounding (Grounding).

**Benchmarks.** For LLMs, we assess instruction following on AlpacaEval (Li et al., 2023), mathematical reasoning on GSM8K (Cobbe et al., 2021) and MATH (Hendrycks et al., 2021), and code generation on HumanEval (Chen et al., 2021) and MBPP (Austin et al., 2021). Unless otherwise noted, we report zero-shot accuracy; for code generation we use pass@1 to measure functional correctness. For MLLMs, we evaluate VQA on VizWiz (Gurari et al., 2018) and GQA (Hudson & Manning, 2019), Geometry on MathVista (Lu et al., 2024) and MATH-Vision (Wang et al., 2024a), Chart on ChartQA (Masry et al., 2022), OCR on TextVQA (Singh et al., 2019) and OCRVQA (Mishra et al., 2019), and Grounding on RefCOCO (Kazemzadeh et al., 2014)/RefCOCO+(Kazemzadeh et al., 2014)/RefCOCOg (Mao et al., 2016). We run the MLLM evaluations using the VLMEvalKit (Duan et al., 2024) and LMMs-Eval (Zhang et al., 2024) toolchains under matched settings to ensure fair comparison. When evaluating AlpacaEval, MathVista, and MATH-Vision, we employ the GPT-4o-mini API to evaluate or extract answers from the output.

**Baselines.** We compare against strong training-free and training-based model-merging baselines: Weight Average (Wortsman et al., 2022), Task Arithmetic (TA) (Ilharco et al., 2022), TIES Merging (Yadav et al., 2023), DARE (Yu et al., 2024), TSV Merging (Gargiulo et al., 2025), Iso-C (Marczak et al., 2025), WUDI Merging (Cheng et al., 2025), WUDI v2 (Wei et al., 2025), and AdaMerging/AdaMerging++ (Yang et al., 2024). Detailed baseline descriptions are provided in Appendix B.

**Implementation details.** For each task, we sample 5–10 unlabeled examples from the training or test sets as the dataset. We first perform Expert Merging, and then apply Expert Merging++ using the learned coefficients. All experiments are repeated five times and results are averaged, using $8\times$ 32G GPUs. For optimizing the Expert Merging/++ objective, we use the AdamW optimizer with a learning rate of $5 \times 10^{-1}$ and weight decay of $1 \times 10^{-5}$, and run 5 epochs with early stopping. The learning rate follows a cosine annealing schedule (Loshchilov & Hutter, 2016) over optimization steps. Additional hyperparameters and details are provided in Appendix B.2.

## 4.2 MAIN RESULTS

Across both the MLLM and LLM backbones, our approach delivers the strongest overall performance while using only unlabeled data for calibration. On InternVL (Table 1), Expert Merging++ attains the best average of 58.45, outperforming the best prior merging method (+1.49 over WUDI v2) and even exceeding supervised Mixture Training (+0.79). The gains are particularly pronounced on grounding benchmarks, where *Expert Merging/++* achieves 80.05/80.53 on RefCOCO, 73.85/74.37 on RefCOCO+, and 79.04/79.31 on RefCOCOg, substantially surpassing all baselines. At the same time, VQA performance remains competitive (e.g., 31.16 on VizWiz, the best in table) and geometry scores are strong (26.32 on MATH-Vision with Expert Merging), indicating that the method effectively balances recognition, reasoning, and localization. While certain OCR-centric metrics (e.g., OCRVQA) favor Mixture Training, the overall average and grounding improvements demonstrate superior cross-domain integration. Additional analysis and results are provided in Appendix C.

Results on Qwen2-VL (Table 2) show the same pattern. Expert Merging++ achieves the best average of 63.63, outperforming the strongest prior merging baseline (+1.00 over WUDI v2) and Mixture Training (+1.40). It sets or matches state-of-the-art results on several domains, including MATH-Vision (44.74) and TextVQA (81.65), while delivering excellent grounding (e.g., 79.00 on RefCOCOg). Although some methods peak on individual tasks (e.g., TA+DARE on ChartQA), they do so at the expense of other domains, whereas *Expert Merging/++* maintains consistently high performance across VQA, geometry, OCR, and grounding, yielding the best overall trade-off. Across InternVL and Qwen2-VL, Expert Merging++ thus consistently attains the highest average over ten evaluation tasks while remaining competitive on each individual benchmark.

---

[1]`https://huggingface.co/mistralai/Mistral-7B-Instruct-v0.1`
[2]`https://huggingface.co/TIGER-Lab/MAmmoTH2-7B`
[3]`https://huggingface.co/Nondzu/Mistral-7B-codealpaca-lora`

Table 1: Main results on InternVL2.5 across multiple tasks.

| Method | VQA | | Geometry | | Chart | OCR | | Grounding | | | Avg. |
|---|---|---|---|---|---|---|---|---|---|---|---|
| | VizWiz | GQA | MathVista | MATH-Vision | ChartQA | TextVQA | OCRVQA | RefCOCO | RefCOCO+ | RefCOCOg | |
| **InternVL2.5-Instruct** | 29.15 | 54.62 | 46.80 | 18.42 | 69.48 | 72.51 | 41.08 | 71.69 | 65.41 | 67.40 | 53.65 |
| VQA Expert | 30.58 | 60.91 | 35.50 | 17.11 | 48.76 | 63.68 | 36.04 | - | - | - | 41.79 |
| Geometry Expert | 13.45 | 32.80 | 55.20 | 25.00 | 51.76 | 56.91 | 35.35 | 24.73 | 19.61 | 23.84 | 33.86 |
| Chart Expert | 20.16 | 40.39 | 23.84 | 10.53 | 69.52 | 54.36 | 34.83 | - | - | - | 36.23 |
| OCR Expert | 12.40 | 22.22 | 23.31 | 10.53 | 36.88 | 73.00 | 54.79 | 73.65 | 68.01 | 69.10 | 44.38 |
| Grounding Expert | 19.09 | 25.88 | 28.91 | 14.47 | 41.32 | 58.39 | **74.87** | 76.67 | 71.35 | 70.09 | 48.10 |
| *Training-Free Model Merging Methods* | | | | | | | | | | | |
| Weight Average | 29.96 | 54.89 | 49.60 | 18.42 | 71.64 | 74.54 | 41.86 | 52.62 | 45.29 | 52.39 | 49.12 |
| Task Arithmetic | 30.67 | 56.34 | 45.36 | 21.05 | 72.88 | 76.26 | 43.39 | 74.90 | 68.15 | 72.75 | 56.17 |
| TIES Merging | 30.63 | 56.48 | 44.50 | 23.68 | 72.28 | 76.29 | 44.01 | 76.01 | 68.45 | 73.65 | 56.59 |
| TA w/ DARE | 30.61 | 56.48 | 48.45 | 21.05 | **73.08** | 76.30 | 43.03 | 74.94 | 68.07 | 73.02 | 56.50 |
| TIES w/ DARE | 30.65 | 56.11 | 43.85 | 27.63 | 72.72 | 76.19 | 43.33 | 75.10 | 68.48 | 73.55 | 56.76 |
| TSV Merging | 31.15 | 56.67 | 52.45 | 28.95 | 70.56 | 75.66 | 45.38 | 65.19 | 58.51 | 59.17 | 54.36 |
| Iso-C | 28.21 | 55.36 | 48.96 | 21.05 | 70.56 | 69.34 | 46.51 | 72.72 | 66.56 | 68.50 | 54.77 |
| *Training-Based Model Merging Methods* | | | | | | | | | | | |
| WUDI Merging | 30.22 | 56.07 | 52.85 | 19.74 | 70.00 | 75.48 | 45.74 | 75.61 | 69.40 | 73.53 | 56.86 |
| WUDI v2 | 31.00 | 57.23 | 52.61 | 21.05 | 65.32 | 76.20 | 45.83 | 76.30 | 69.91 | 74.16 | 56.96 |
| AdaMerging | 31.06 | 57.20 | 55.10 | 25.00 | 62.32 | 76.20 | 44.76 | 75.30 | 68.42 | 73.22 | 56.85 |
| AdaMerging++ | 30.94 | 57.18 | 51.77 | 19.74 | 66.20 | 76.02 | 45.93 | 76.45 | 70.47 | 74.41 | 56.91 |
| Mixture Training | 29.79 | **61.33** | 52.83 | 23.68 | 70.32 | 72.96 | 60.25 | 72.06 | 65.93 | 67.46 | 57.66 |
| Expert Merging | **31.16** | 56.77 | 47.71 | 26.32 | 65.28 | 75.70 | 45.31 | 80.05 | 73.85 | 79.04 | 58.11 |
| Expert Merging++ | 31.14 | 56.53 | 52.35 | 22.37 | 65.92 | 76.06 | 46.00 | **80.53** | 74.37 | 79.31 | 58.45 |

Table 2: Main results on Qwen2-VL across multiple tasks.

| Method | VQA | | Geometry | | Chart | OCR | | Grounding | | | Avg. |
|---|---|---|---|---|---|---|---|---|---|---|---|
| | VizWiz | GQA | MathVista | MATH-Vision | ChartQA | TextVQA | OCRVQA | RefCOCO | RefCOCO+ | RefCOCOg | |
| **Qwen2-VL-Base** | 5.52 | 5.39 | 47.85 | 23.68 | 0.36 | 20.22 | 1.07 | 45.32 | 37.55 | 31.26 | 21.82 |
| VQA Expert | 41.38 | **62.60** | 33.71 | 28.94 | 66.56 | 80.21 | 55.33 | 39.31 | 32.71 | 38.01 | 47.88 |
| Geometry Expert | 35.57 | 44.63 | 42.50 | 28.95 | 14.56 | 73.95 | 45.96 | 5.57 | 2.31 | 3.90 | 29.79 |
| Chart Expert | 38.58 | 24.24 | 49.28 | 32.89 | 61.08 | 79.75 | 63.67 | 46.28 | 36.67 | 34.06 | 46.65 |
| OCR Expert | 28.38 | 37.53 | 31.81 | 13.16 | 57.40 | 70.50 | 64.68 | 0.59 | 0.46 | 0.26 | 30.48 |
| Grounding Expert | 38.60 | 32.92 | 36.17 | 19.74 | 18.08 | 75.05 | 48.27 | 72.14 | 65.33 | 66.48 | 47.28 |
| *Training-Free Model Merging Methods* | | | | | | | | | | | |
| Weight Average | 41.47 | 57.33 | 50.21 | 34.21 | 59.56 | 81.09 | 57.85 | 80.72 | 65.37 | 77.68 | 60.55 |
| Task Arithmetic | 40.52 | 62.31 | 40.36 | 26.31 | 79.67 | 81.09 | 59.50 | 75.96 | 61.33 | 75.85 | 60.29 |
| TIES Merging | 41.38 | 59.08 | 46.87 | 34.21 | 67.24 | 81.42 | 58.53 | 80.63 | 65.36 | 77.65 | 61.24 |
| TA w/ DARE | 40.64 | 62.38 | 40.67 | 26.31 | **79.76** | 81.04 | 59.34 | 75.83 | 61.41 | 75.80 | 60.32 |
| TIES w/ DARE | 41.63 | 59.96 | 45.72 | 35.53 | 70.68 | 81.53 | 59.63 | 80.73 | 65.65 | 77.77 | 61.88 |
| TSV Merging | 41.43 | 57.31 | 51.05 | 34.21 | 59.44 | 81.25 | 57.81 | 80.71 | 65.34 | 77.76 | 60.63 |
| Iso-C | 12.31 | 13.44 | 39.96 | 27.63 | 2.80 | 30.05 | 6.12 | 53.68 | 38.96 | 41.90 | 26.69 |
| *Training-Based Model Merging Methods* | | | | | | | | | | | |
| WUDI Merging | 37.19 | 56.45 | 42.96 | 27.63 | 67.84 | 79.92 | **65.56** | 76.25 | 60.72 | 71.99 | 58.65 |
| WUDI v2 | 41.13 | 61.17 | 46.98 | 38.16 | 74.64 | 79.54 | 60.03 | 80.43 | 65.85 | 78.42 | 62.63 |
| AdaMerging | 41.83 | 60.28 | 49.08 | 35.53 | 71.68 | 81.52 | 60.22 | 53.31 | 47.27 | 57.62 | 55.83 |
| AdaMerging++ | 41.67 | 60.09 | 47.82 | 35.53 | 69.76 | 81.59 | 60.42 | 80.57 | 65.98 | 78.74 | 62.22 |
| Mixture Training | **44.09** | 62.18 | 46.02 | 19.73 | 70.04 | 78.38 | 65.42 | **82.89** | **77.87** | 75.63 | 62.23 |
| Expert Merging | 41.54 | 60.51 | 48.80 | 42.11 | 73.00 | 81.41 | 60.19 | 80.64 | 66.15 | **79.02** | 63.34 |
| Expert Merging++ | 41.49 | 60.42 | 49.33 | **44.74** | 72.60 | **81.65** | 60.19 | 80.71 | 66.21 | 79.00 | 63.63 |

For Mistral-7B (Table 3), Expert Merging++ attains the highest average (48.71), improving over strong baselines such as WUDI v2 (+3.23) and AdaMerging++ (+3.61). It achieves the best Code score among merged models (53.66 on HumanEval), with strong performance on AlpacaEval (75.51) and GSM8K (59.44). Notably, Expert Merging attains the highest Chat score overall (77.84 on AlpacaEval). Our merged models approach or match specialist experts within their domains while consolidating all capabilities into a single model, validating the effectiveness of our coefficient learning and alignment objectives. Together, these results support our key claim: aligning both hidden states and logits with per-layer coefficients yields a robust, training-efficient path to unify domain experts. The chunk-wise coefficients of Expert Merging++ (guided by the importance analysis in Section 4.3) further improve cross-domain balance with negligible overhead, leading to state-of-the-art averages across backbones without supervised finetuning.

## 4.3 LAYER-WISE IMPORTANCE ANALYSIS

Merging performance is not uniform across layers or depths. To investigate this heterogeneity, we examined the coefficient magnitudes and parameter sizes of each layer across different component types (attention projections Q/K/V/O and MLP Gate/Up/Down) and depths (early, middle, late). We

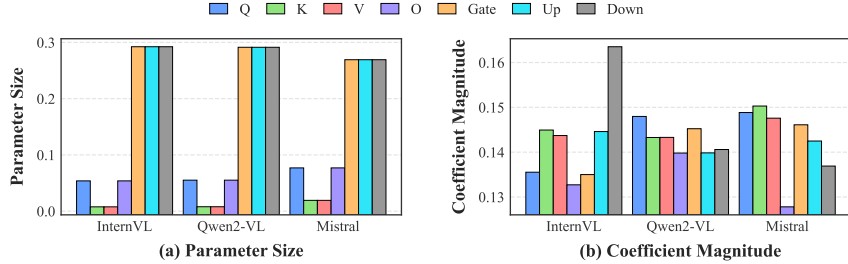

Figure 3: Model-wise parameter size and learned coefficient across backbones after normalizing. Full stage-wise trends for panel (b) are provided in Appendix (Figure 5).

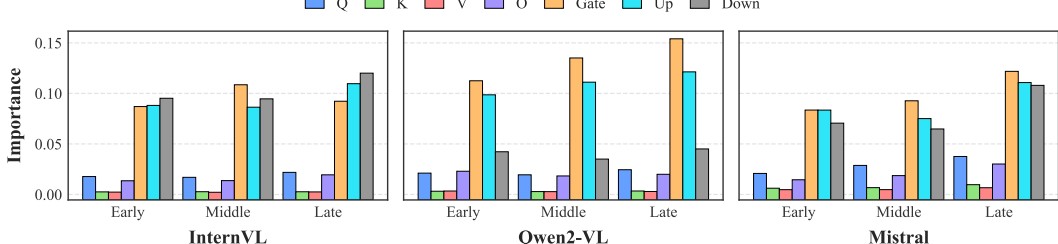

Figure 4: Layer importance by stage and submodule across backbones.

then combined these statistics with the task vector weights to derive a per-layer importance score, quantifying each layer's contribution to the final merged model.

**Parameter size across backbones.** Despite architectural differences, the distribution of parameters by submodule type is highly consistent (panel (a) of Figure 3). In Qwen and InternVL, the MLP dominates capacity: Gate/Up/Down together account for roughly 87% of the parameters, while attention uses the remaining 13%. Mistral has a slightly larger attention parameters (about 19%) but the same qualitative pattern. This structure suggests that most of the representational capacity resides in the MLP, with Q/O providing relatively smaller but non-negligible attention-side capacity.

**Learned coefficient magnitude.** After normalizing the learned coefficients within each model, we aggregate them across submodule to obtain panel (b) of Figure 3. The magnitudes are fairly balanced overall, but the leading submodule varies by backbone: Down is largest for InternVL, Q for Qwen2-VL, and K for Mistral. The O projection is consistently the smallest across all three. Notably, Down exhibits the greatest variance; it is highest on InternVL while being lowest on Mistral. This indicates backbone-specific allocation rather than a universal ranking. This pattern contrasts with panel (a), where MLP (Gate/Up/Down) clearly dominates by parameter size.

**Importance: where capacity actually matters.** As described in Section 3.3, we compute layer importance by stage and submodule

Table 3: Main results on Mistral across multiple tasks.

| Method | Chat | MATH | | Code | | |
|---|---|---|---|---|---|---|
| | Alpaca Eval | GSM8k | Math | HumanEval | MBPP | Avg. |
| **Mistral-7B-v0.1** | 4.87 | 13.57 | 5.10 | 30.49 | 40.80 | 18.97 |
| **Chat Expert** | 71.37 | 41.09 | 7.26 | 35.37 | 32.20 | 37.46 |
| **Math Expert** | 53.18 | **63.38** | **25.22** | 20.73 | 32.40 | 38.98 |
| **Code Expert** | 69.90 | 30.78 | 4.84 | **54.88** | 38.20 | 39.72 |
| *Training-Free Model Merging Methods* | | | | | | |
| **Task Arithmetic** | 74.18 | 52.69 | 13.16 | 45.12 | 26.00 | 42.23 |
| **TIES Merging** | 71.66 | 53.68 | 13.86 | 38.41 | 40.00 | 43.52 |
| **TA w/ DARE** | 74.92 | 56.71 | 13.22 | 43.29 | **43.40** | 42.71 |
| **TIES w/ DARE** | 76.33 | 52.08 | 12.00 | 34.76 | **43.40** | 43.71 |
| **TSV Merging** | 60.61 | 47.38 | 11.18 | 37.20 | 42.20 | 39.71 |
| **Iso-C** | 69.76 | 52.39 | 12.82 | 32.32 | 40.60 | 41.58 |
| *Training-Based Model Merging Methods* | | | | | | |
| **WUDI Merging** | 64.93 | 53.37 | 14.04 | 32.93 | **43.40** | 41.73 |
| **WUDI v2** | 73.84 | 59.06 | 14.88 | 39.02 | 40.60 | 45.48 |
| **AdaMerging** | 75.94 | 54.81 | 12.24 | 43.29 | 39.00 | 45.06 |
| **AdaMerging++** | 75.94 | 54.81 | 12.24 | 43.29 | 39.20 | 45.10 |
| **Expert Merging** | **77.84** | 58.68 | 13.88 | 46.34 | 40.40 | 47.43 |
| **Expert Merging++** | 75.51 | 59.44 | 13.94 | 53.66 | 41.00 | **48.71** |

across backbones, with results shown in Figure 4. First, MLP components dominate: Gate and Up consistently rank as the two largest contributors, followed by Down; attention K and V contribute the least; Q and O fall in between. Second, importance concentrates in the late-stage third of the network for all types, with monotonic or near-monotonic increases from early to late blocks (particularly pronounced for Mistral). Third, backbone-specific nuances remain, InternVL's Down-projection is the single largest contributor, whereas Mistral distributes MLP importance more evenly between Gate and Up, but the overarching ranking is consistent.

**Takeaways for Expert Merging++.** The analysis reveals substantial variation across submodules and depths in different architectures, indicating that the prior practice of assigning a single coeffi-

cient per layer overlooks critical inter-layer heterogeneity. This observation motivates Expert Merging++, which replaces the layer-wise coefficient with chunk-wise coefficients, enabling finer-grained control in high-importance layers. In particular, high-importance layers exhibit larger merged task vectors and influence more parameters, explaining why chunking only these layers recovers nearly all the benefits of dense chunking, whereas chunking low-importance layers yields little gain.

## 4.4 ABLATION STUDIES

We study how each component of Expert Merging and Expert Merging++ contributes to performance across the two MLLM backbones and the Mistral LLM. For brevity, we report a consolidated summary in Table 4, and defer the full per-benchmark ablation tables to Appendix C.3.

**Alignment objectives.** Removing either alignment term degrades performance across all backbones, confirming that matching both internal representations and predictive distributions is beneficial. On InternVL and Mistral, logit alignment accounts for the larger gains ($-3.39$ and $-3.81$ average without it), while hidden-state alignment provides additional improvements. On Qwen2-VL, the two terms contribute com-

Table 4: Ablations on key design choices.

| Variant | InternVL | Qwen2-VL | Mistral |
|---|---|---|---|
| Expert Merging | 58.11 | 63.34 | 47.43 |
|   w/o hidden alignment | 57.75 | 62.43 | 47.10 |
|   w/o logit alignment | 54.72 | 62.71 | 43.62 |
|   w/o coefficient regularization | 47.57 | 62.17 | 40.81 |
|   w/o task trade-offs | 57.71 | 62.69 | 46.80 |
| Expert Merging++ | **58.45** | **63.63** | **48.71** |
|   w/o chunking | 58.11 | 63.34 | 47.43 |
|   chunk all layers (2x) | 56.76 | 62.61 | 46.82 |

parably. Qualitatively, hidden alignment stabilizes cross-domain behavior (notably OCR and grounding on InternVL), while logit knowledge aligning brings task-faithful outputs.

**Coefficient regularization.** The coefficient regularizer is critical for stability. Without it, averages drop markedly ($-10.54$ on InternVL, $-6.62$ on Mistral, $-1.17$ on Qwen2-VL), accompanied by pronounced degradation on grounding metrics (e.g., RefCOCO/+/g on InternVL). This supports our design in Section 3: regularization curbs coefficient drift during unsupervised alignment and prevents catastrophic forgetting in high-capacity, late layers.

**Controllable task trade-offs.** Removing explicit task weights consistently reduces averages ($-0.4$ to $-0.7$). Beyond the modest aggregate gains, the weights enable practical control over cross-domain balance, aligning with our goal of exposing interpretable knobs for deployment.

**Chunk-wise coefficients.** Expert Merging++ extends layer-wise coefficients (Expert Merging) by assigning chunk-wise coefficients, yielding further improvements: +0.34 on InternVL, +0.29 on Qwen2-VL, and +1.28 on Mistral. These gains concentrate in regions where our importance analysis (Section 4.3) indicates capacity matters most, namely late-stage MLP blocks. For example, chunking substantially boosts code generation on Mistral (HumanEval +7.3 points) while preserving or slightly improving other domains. In contrast, indiscriminately chunking all layers (2× coefficients) consistently degrades performance across backbones. This underscores that targeted capacity allocation, rather than uniform over-parameterization, is crucial for addressing inter-layer heterogeneity, and further validates both our importance metric and the chunk-wise design of Expert Merging++.

**Summary.** The ablations validate our core choices: (1) alignment losses both help (logits stronger on InternVL/Mistral, hidden states aid robustness); (2) coefficient regularization is critical for stability; (3) task weights enable controllable trade-offs; and (4) chunk-wise coefficients in Expert Merging++ deliver further gains. Together, these components yield the best balance between accuracy and stability and deliver consistent cross-domain improvements. Combined with the main tables, these component wise degradations support our claim that the gains of Expert Merging and Expert Merging++ come from the hidden state and logit alignment objectives plus importance guided chunking rather than incidental engineering choices.

## 5 CONCLUSION AND FUTURE WORK

We introduce Expert Merging, a label-free, training-light approach that learns layer-wise coefficients to align hidden states and logits, and Expert Merging++ which adds importance-guided, chunk-wise capacity. Across InternVL, Qwen2-VL, and Mistral, they consistently surpass training-free and prior training-based baselines; Expert Merging++ provides further gains and in some cases exceeds supervised Mixture Training, with ablations confirming each design choice. In future work, we plan to further explore model merging solutions for heterogeneous models.

REPRODUCIBILITY STATEMENT

Algorithmic details, coefficient parameterization, and the alignment objectives are specified in Section 3 (with Figures 1 and 2). The complete experimental setup—including backbones, expert checkpoints, benchmarks, evaluation toolchains, and baselines—is described in Section 4. Hyper-parameters, data sampling protocols, coefficient initialization/regularization, chunking budgets, and compute settings are documented in Appendix B.2, and baseline descriptions and evaluation settings are summarized in Appendix B. We report consolidated ablations in Table 4 and provide the full per-benchmark ablations in Appendix C.3. For evaluation consistency, we rely on public toolchains (VLMEvalKit and LMMs-Eval) under matched settings as cited in Section 4. Our code is available at `https://github.com/Littleor/ExpertMerging` to further facilitate reproduction of our method.

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

## A USE OF LARGE LANGUAGE MODELS

We used large language models (LLMs) to aid and polish the writing of this manuscript. Concretely, we used LLMs to suggest improvements to grammar, phrasing, and clarity for select passages.

## B EXPERIMENT SETTING

### B.1 BASELINES

We compare against the following baselines:

- **Weight Averaging** (Wortsman et al., 2022): simply averages the weights (or deltas) of models fine-tuned on different tasks.

- **Task Arithmetic (TA)** (Ilharco et al., 2022): define task vectors $\tau_k = \theta_k - \theta_{\text{base}}$, and the merged model follows the notation in Section 3.1 as $\theta_{\text{TA}} = \theta_{\text{base}} + \sum_{k=1}^{K} \lambda_k \tau_k$ (with predefined $\{\lambda_k\}$). **DARE** (Yu et al., 2024) randomly drops redundant task vectors and rescales the remaining ones to mitigate interference.

- **TIES Merging** (Yadav et al., 2023): a sparsification-based pipeline with Trim, Elect Sign, and Disjoint Merge to produce a merged vector $\tau_m$; the final model is $\theta_{\text{base}} + \lambda \tau_m$ with $\lambda$ tuned; we also evaluate **TIES+DARE**.

- **TSV Merging** (Gargiulo et al., 2025): an SVD-based method that measures singular task interference and reduces it via decorrelation formulated as an orthogonal Procrustes problem.

- **Iso-C** (Marczak et al., 2025): isotropic merging that flattens the singular value spectrum of task matrices and improves alignment between singular components.

- **WUDI Merging** (Cheng et al., 2025): optimization-based merging that treats task vectors as spanning an approximate linear subspace and minimizes layer-wise interference between the merged vector $\tau_{m,\ell}$ and each task vector $\tau_{i,\ell}$ using only task vectors (no labels).

- **WUDI v2** (Wei et al., 2025): adds an SVD-based component to decompose task matrices, yielding stronger multi-domain performance.

- **AdaMerging** (Yang et al., 2024): training-based approach that learns small, typically per-layer coefficients on unlabeled calibration data to combine experts.

- **AdaMerging++** (Yang et al., 2024): a TIES-based variant that learns per-layer coefficients for each task vector *within the TIES-Merging pipeline*, improving robustness to conflicts.

- **Mixture Training**: supervised multi-task finetuning on the union of data, serving as an upper bound reference when available.

**Analysis.** Across backbones (Tables 1, 2, and 3), training-free methods exhibit clear trade-offs. TA/DARE variants often peak on a few domains (e.g., ChartQA) but underperform on grounding or geometry tasks, while sparsification- and SVD-based methods (TIES/TSV/Iso-C) reduce conflicts yet still lag on cross-domain balance. Optimization-oriented WUDI/WUDI v2 deliver stronger macro averages among training-free baselines but remain below *Expert Merging/++*, especially on grounding (MLLMs) and code (LLMs). Notably, Mixture Training is not a strict upper bound in our setting: *Expert Merging++* surpasses it on macro average for both InternVL and Qwen2-VL, echoing the main text that careful, unlabeled calibration can rival or exceed supervised multi-task fine-tuning.

### B.2 IMPLEMENTATION DETAILS

**Backbones and experts.** We consider both LLMs and MLLMs. For LLMs, the backbone is Mistral-7B-v0.1 with three domain experts: instruction following (Chat), mathematical reasoning (Math), and code generation (Code). For MLLMs, following prior multimodal merging work (Wei et al., 2025), we use InternVL2.5-1B-Instruct and Qwen2-VL-7B-Base as backbones, with five capability experts: VQA, Geometry, Chart, OCR, and Grounding. Expert checkpoints follow the sources referenced in Section 4.

**Hyperparameters.**  We report the key hyperparameters used; other settings follow defaults and are omitted due to space.

- **Train samples per task ($N$).** InternVL2.5: **5**; Mistral-7B-v0.1: **5**; Qwen2-VL-7B-Base: **10**.

- **Initialization ($\bar{\alpha}$ prior).** InternVL2.5: **0.1**; Mistral-7B-v0.1: **0.3**; Qwen2-VL-7B-Base: **0.3**.

- **Hidden alignment layers ($\mathcal{S}$).** For all models, we supervise only the *middle* Transformer layer's hidden state, i.e., $|\mathcal{S}|=1$ with layer index $\lfloor L/2 \rfloor$.

- **Coefficient regularization weight ($\gamma$).** InternVL2.5: **5**; Mistral-7B-v0.1: **0.8**; Qwen2-VL-7B-Base: **0.8**.

- **Total coefficient budget ($B$).** Set to $0.9$–$1.2 \times L$ to reduce trainable parameters and encourage sparsity.

- **Allocation steepness ($\kappa$).** Set to **1.2** for each model to accentuate differences in layer importance.

## C  EXPERIMENT RESULTS

### C.1  LAYER-WISE IMPORTANCE ANALYSIS

This section provides the full stage-wise (early/middle/late) breakdown of learned coefficients for each submodule and backbone, complementing the model-wise aggregation in Figure 3b.

**Learned coefficients across depth.**  Coefficients learned by *Expert Merging* are distinctly non-uniform over depth as Figure 5 shown. Across all three backbones, late-stage blocks receive larger shares than early ones for both attention and MLP. For example, on Mistral the Q and Gate shares grow from about ∼0.04 (early) to ∼0.06 (late). InternVL shows a pronounced late-stage skew on the MLP down-projection, while Qwen exhibits a milder early-to-late increase. Averaged over types, model-wise coefficient shares remain relatively even (roughly 0.13–0.16 per type), suggesting that raw coefficient fractions alone weakly indicate capacity hot spots unless viewed alongside parameterization across depth.

**Interpretation.**  The late-layer emphasis aligns with the role of deeper blocks in consolidating task-specific signals (e.g., cross-modal grounding in MLLMs and semantic control in LLMs). This observation motivated the chunk-wise refinement in *Expert Merging++*, which reallocates more capacity primarily within later regions. Consistently, Appendix C.3 shows targeted chunking improves grounding/OCR on InternVL and boosts code/math on Mistral with minimal impact on VQA.

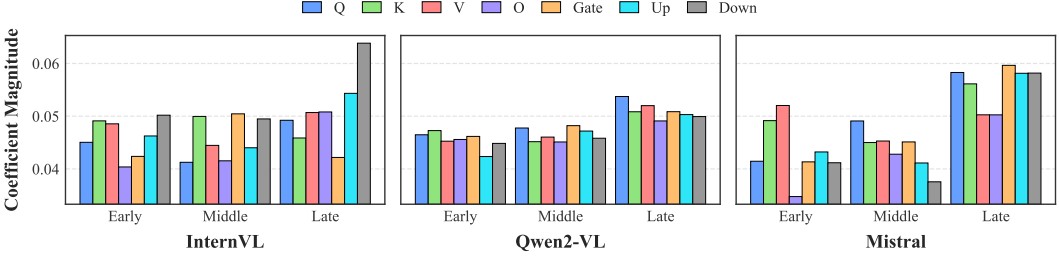

Figure 5: Learned coefficient share across depth (early/middle/late) for each submodule. Late-stage blocks consistently receive larger shares for both attention and MLP; InternVL shows a stronger late-stage skew on the MLP down-projection, while Qwen's shift is milder.

### C.2  TASK-VECTOR COSINE SIMILARITY ANALYSIS

To better understand interactions among experts, we compute cosine similarity between task vectors $\tau_k = \theta_k - \theta_{\text{base}}$. Figure 6 reports heatmaps for InternVL, Qwen2-VL, and Mistral.

Table 5: Ablation studies on InternVL2.5. We measure the effect of each component in Expert Merging and the chunking strategy in Expert Merging++.

| | VQA | | Geometry | | Chart | OCR | | Grounding | | | |
|---|---|---|---|---|---|---|---|---|---|---|---|
| Variant | VizWiz | GQA | MathVista | MATH-Vision | ChartQA | TextQA | OCRVQA | RefCOCO | RefCOCO+ | RefCOCOg | Avg. |
| **Expert Merging** | 31.16 | 56.77 | 47.71 | 26.32 | 65.28 | 75.70 | 45.31 | 80.05 | 73.85 | 79.04 | 58.11 |
| w/o hidden alignment | 31.07 | 56.81 | 49.90 | **27.63** | 64.52 | 75.55 | 45.15 | 78.06 | 71.92 | 76.98 | 57.75 |
| w/o logit alignment | **31.53** | **57.08** | 48.82 | 22.37 | **67.16** | 75.82 | 44.76 | 69.81 | 63.05 | 66.83 | 54.72 |
| w/o task trade-offs | 31.16 | 56.72 | 51.17 | 26.32 | 64.60 | 75.52 | 45.05 | 78.12 | 71.65 | 76.82 | 57.71 |
| w/o coefficient regularization | 29.63 | 53.57 | 52.11 | 22.37 | 65.08 | 73.13 | 43.36 | 46.68 | 40.96 | 48.86 | 47.57 |
| **Expert Merging++** | 31.14 | 56.53 | **52.35** | 22.37 | 65.92 | **76.06** | **46.00** | 80.53 | 74.37 | 79.31 | **58.45** |
| no chunking | 31.16 | 56.77 | 47.71 | 26.32 | 65.28 | 75.70 | 45.31 | 80.05 | 73.85 | 79.04 | 58.11 |
| chunk all layers (2x) | 31.03 | 56.86 | 50.12 | 21.05 | 63.40 | 75.57 | 45.05 | 77.40 | 71.02 | 76.12 | 56.76 |

On InternVL, most off diagonal entries are close to zero, with the largest similarity between OCR and Grounding ($\approx 0.14$). This suggests that the visual experts are largely complementary and that conflicts arise mainly from a few pairs (OCR and Grounding) with modest overlap. On Qwen2-VL, similarities are more uniformly positive ($\approx 0.05$ to $0.13$), especially among OCR, VQA, and Chart, indicating shared structure among text heavy visual tasks, while Geometry and Grounding remain less aligned with them. For Mistral, the Chat and Code experts show a relatively high similarity ($\approx 0.42$), whereas Math is almost orthogonal to both. This matches our observation that code and instruction tuning are more synergistic than math, yet the merged model retains strong performance across all three domains due to the behavior space alignment objective and coefficient regularization.

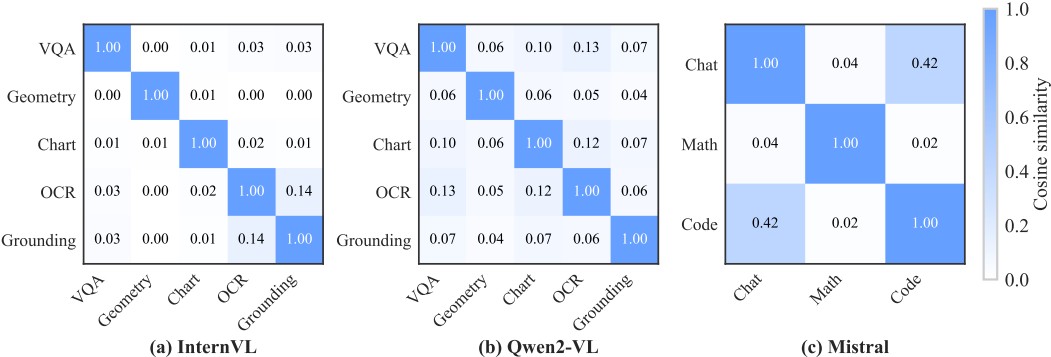

| (a) InternVL | (b) Qwen2-VL | (c) Mistral |

Figure 6: Task-vector cosine similarity heatmaps for (a) InternVL, (b) Qwen2-VL, and (c) Mistral. Darker blocks indicate higher similarity.

## C.3 ABLATION STUDIES

This section provides the complete ablation results for each backbone, expanding the summarized findings reported in Table 4. See Tables 5, 6, and 7 for detailed results.

**Summary.** Three consistent trends emerge across backbones: (i) *logit alignment is essential*—removing it causes the largest average drops and notably hurts geometry/grounding on MLLMs and code on LLMs; (ii) *coefficient regularization is critical for stability*—without it, performance collapses on InternVL and Mistral, indicating over-allocation and interference; and (iii) *chunking helps when guided by importance*—the targeted chunk-wise variant in *Expert Merging++* yields macro-average gains, whereas naively chunking all layers degrades multiple domains.

*InternVL2.5.* Table 5 shows that, relative to *Expert Merging*, removing hidden alignment slightly raises geometry (e.g., MathVista/MATH-Vision) but reduces grounding by 2–3 points, yielding a lower average—indicating hidden features help localization. Removing logit alignment severely hurts geometry and grounding (while sometimes inflating ChartQA), confirming logits are indispensable for cross-domain consistency. Eliminating coefficient regularization leads to a dramatic collapse, especially on grounding benchmarks. Guided chunking in *Expert Merging++* improves OCR and grounding and achieves the best average, while chunking all layers uniformly harms geometry and grounding.

Table 6: Ablation studies on Qwen2-VL. We report the impact of each loss component and chunking strategy.

| | VQA | | Geometry | | Chart | OCR | | Grounding | | | |
|---|---|---|---|---|---|---|---|---|---|---|---|
| Variant | VizWiz | GQA | MathVista | MATH-Vision | ChartQA | TextVQA | OCRVQA | RefCOCO | RefCOCO+ | RefCOCOg | Avg. |
| **Expert Merging** | 41.54 | 60.51 | 48.80 | 42.11 | **73.00** | 81.41 | 60.19 | 80.64 | 66.15 | **79.02** | 63.34 |
| w/o hidden alignment | 41.64 | 60.35 | 47.66 | 34.21 | 72.84 | 81.59 | **60.38** | 80.68 | 66.05 | 78.87 | 62.43 |
| w/o logit alignment | **41.80** | 60.38 | 47.82 | 38.16 | 72.04 | 81.44 | 60.16 | 80.64 | 65.89 | 78.76 | 62.71 |
| w/o task trade-offs | 41.56 | 60.53 | 48.38 | 36.84 | 72.80 | 81.49 | 59.96 | 80.51 | 66.00 | 78.82 | 62.69 |
| w/o coefficient regularization | 41.67 | **60.82** | **49.63** | 36.84 | 72.20 | 81.59 | 60.12 | 78.33 | 63.90 | 76.59 | 62.17 |
| **Expert Merging++** | 41.49 | 60.42 | 49.33 | **44.74** | 72.60 | **81.65** | 60.19 | 80.71 | 66.21 | 79.00 | **63.63** |
| no chunking | 41.54 | 60.51 | 48.80 | 42.11 | **73.00** | 81.41 | 60.19 | 80.64 | 66.15 | **79.02** | 63.34 |
| chunk all layers (2x) | 41.47 | 60.55 | 46.41 | 38.16 | 72.72 | 81.45 | 60.12 | 80.49 | 65.87 | 78.90 | 62.61 |

Table 7: Ablation studies on Mistral. We evaluate the effect of each loss component and chunking choice on text benchmarks.

| | Chat | MATH | | Code | | |
|---|---|---|---|---|---|---|
| Variant | AlpacaEval | GSM8K | MATH | HumanEval | MBPP | Avg. |
| **Expert Merging** | 77.84 | 58.68 | 13.88 | 46.34 | 40.40 | 47.43 |
| w/o hidden alignment | 77.99 | 57.09 | 13.92 | 45.12 | 41.40 | 47.10 |
| w/o logit alignment | 69.68 | 56.10 | 13.32 | 38.41 | 40.60 | 43.62 |
| w/o task trade-offs | **78.95** | 54.81 | 13.50 | 45.12 | **41.60** | 46.80 |
| w/o coefficient regularization | 65.32 | 52.01 | 12.08 | 39.02 | 35.60 | 40.81 |
| **Expert Merging++** | 75.51 | **59.44** | **13.94** | **53.66** | 41.00 | **48.71** |
| no chunking | 77.84 | 58.68 | 13.88 | 46.34 | 40.40 | 47.43 |
| chunk all layers (2x) | 77.84 | 56.79 | 13.74 | 44.51 | 41.20 | 46.82 |

*Qwen2-VL.* Table 6 shows that sensitivity is milder: removing hidden or logit alignment changes the average only modestly, though MATH-Vision degrades without logits. Dropping regularization shifts scores unevenly (e.g., slight gains on GQA/MathVista but clear declines in grounding), underscoring the stability–variance trade-off. Importance-guided chunking improves MATH-Vision and the macro average; indiscriminate chunk-all-layers again underperforms.

*Mistral-7B.* Table 7 shows that removing logit alignment yields the largest drop among ablations, impacting both instruction following and code. The coefficient regularizer is likewise crucial: without it, average performance falls sharply. *Expert Merging++* delivers the best average, with gains dominated by a large improvement on HumanEval and consistent lifts on GSM8K/MBPP, while AlpacaEval remains strong. Uniform chunking degrades results relative to targeted chunking.

## C.4 REGULARIZATION SENSITIVITY

We evaluate the effect of the coefficient regularization strength on *Expert Merging*, keeping all other settings fixed. Metrics and datasets match the main ablations.

**Summary.** Across backbones, moderate coefficient regularization yields the best macro averages: InternVL2.5 prefers a stronger setting ($\gamma \approx 5$), while Qwen2-VL and Mistral-7B favor smaller values ($\gamma \approx 0.8$). Gains come primarily from grounding (MLLMs) and code (LLMs), with mild trade-offs on Chart/geometry at very large $\gamma$. This behavior matches the role of the coefficient regularizer in curbing interference and preventing over-allocation. See Tables 8, 9, and 10.

**Backbone-specific results.** For InternVL2.5, Table 8 shows the average rises from 43.49 at $\gamma{=}0$ to 56.61 at $\gamma{=}5.0$ (best), then plateaus (56.42 at $\gamma{=}8.0$; 55.93 at $\gamma{=}10.0$). The gain is driven largely by grounding: RefCOCO/RefCOCO+/RefCOCOg improve from 32.46/29.06/40.97 to 79.08/72.91/77.70 at $\gamma{=}5.0$ (79.63/73.42/78.00 at $\gamma{=}8.0$). ChartQA and MATH-Vision favor smaller $\gamma$ (66.76 at 0.6; 26.32 at 0.8), while VQA/OCR change little. Overall, $\gamma \approx 5$ is the best trade-off.

For Qwen2-VL, Table 9 shows sensitivity is mild: the average peaks at 63.34 for $\gamma{=}0.8$ and remains within about 0.5 over $\gamma \in [0, 10]$. MATH-Vision benefits from moderate regularization (42.11 at 0.8 vs. 39.47 at 0), whereas VQA/OCR are nearly unchanged and grounding remains stable around 80.6/66.2/79.0. A smaller default ($\gamma \approx 0.8$) is appropriate.

Table 8: Regularization sensitivity on InternVL2.5 for Expert Merging. Rows vary the coefficient regularization strength.

| | VQA | | Geometry | | Chart | OCR | | Grounding | | | |
|---|---|---|---|---|---|---|---|---|---|---|---|
| $\gamma$ | VizWiz | GQA | MathVista | MATH-Vision | ChartQA | TextVQA | OCRVQA | RefCOCO | RefCOCO+ | RefCOCOg | Avg. |
| $\gamma$=0 | 29.15 | 51.06 | **51.93** | 19.74 | 65.64 | 72.52 | 42.42 | 32.46 | 29.06 | 40.97 | 43.49 |
| $\gamma$=0.2 | 29.75 | 52.97 | 51.04 | 21.05 | 65.96 | 73.47 | 43.39 | 36.85 | 32.28 | 41.22 | 44.79 |
| $\gamma$=0.4 | 30.16 | 54.41 | 50.88 | 22.37 | 66.44 | 73.94 | 44.08 | 38.02 | 32.85 | 38.54 | 45.16 |
| $\gamma$=0.6 | 30.49 | 55.12 | 49.47 | 22.37 | **66.76** | 74.43 | 44.69 | 40.26 | 34.15 | 37.03 | 45.47 |
| $\gamma$=0.8 | 30.63 | 55.52 | 49.98 | **26.32** | 66.16 | 74.87 | 45.05 | 44.13 | 38.03 | 39.97 | 47.06 |
| $\gamma$=0.9 | 30.83 | 55.72 | 48.19 | 25.00 | 65.76 | 74.92 | 45.38 | 46.68 | 40.46 | 41.85 | 47.47 |
| $\gamma$=1.0 | 30.96 | 55.92 | 47.77 | 21.05 | 65.24 | 75.04 | 45.38 | 49.32 | 43.21 | 44.83 | 47.87 |
| $\gamma$=1.5 | 31.15 | 56.12 | 46.45 | 21.05 | 65.08 | 75.42 | 45.05 | 62.61 | 56.18 | 57.41 | 51.65 |
| $\gamma$=2.0 | 31.23 | 56.27 | 47.73 | 21.05 | 64.16 | 75.76 | 44.95 | 71.77 | 65.62 | 68.14 | 54.66 |
| $\gamma$=3.0 | **31.26** | 56.29 | 49.12 | 18.42 | 62.64 | **75.89** | 45.31 | 77.97 | 71.67 | 75.74 | 56.43 |
| $\gamma$=5.0 | 31.05 | 56.49 | 47.71 | 19.74 | 60.12 | 75.87 | 45.44 | 79.08 | 72.91 | 77.70 | **56.61** |
| $\gamma$=8.0 | 30.94 | 56.70 | 47.29 | 18.42 | 58.44 | 75.74 | **45.64** | **79.63** | 73.42 | 78.00 | 56.42 |
| $\gamma$=10.0 | 31.07 | **56.85** | 43.96 | 17.11 | 58.24 | 75.53 | 45.57 | 79.50 | **73.42** | **78.08** | 55.93 |

Table 9: Regularization sensitivity on Qwen2-VL for Expert Merging. Rows vary the coefficient regularization strength.

| | VQA | | Geometry | | Chart | OCR | | Grounding | | | |
|---|---|---|---|---|---|---|---|---|---|---|---|
| $\gamma$ | VizWiz | GQA | MathVista | MATH-Vision | ChartQA | TextVQA | OCRVQA | RefCOCO | RefCOCO+ | RefCOCOg | Avg. |
| $\gamma$=0 | 41.48 | 60.51 | 48.80 | 39.47 | 73.24 | 81.39 | 59.99 | 80.59 | 66.17 | 78.93 | 63.06 |
| $\gamma$=0.2 | 41.45 | 60.42 | 47.96 | 39.47 | **73.36** | 81.40 | 59.99 | 80.59 | 66.06 | 79.01 | 62.97 |
| $\gamma$=0.4 | 41.39 | 60.47 | **49.22** | 39.47 | 73.20 | 81.39 | 60.06 | 80.55 | 66.13 | 78.91 | 63.08 |
| $\gamma$=0.6 | 41.43 | **60.52** | **49.22** | 39.47 | 73.32 | 81.34 | 60.06 | 80.57 | **66.24** | 78.94 | 63.11 |
| $\gamma$=0.8 | **41.54** | 60.51 | 48.80 | **42.11** | 73.00 | 81.41 | **60.19** | **80.64** | 66.15 | **79.02** | **63.34** |
| $\gamma$=0.9 | 41.39 | 60.47 | 48.38 | 39.47 | 73.12 | 81.32 | 60.16 | 80.56 | 66.16 | 78.91 | 62.99 |
| $\gamma$=1.0 | 41.42 | 60.49 | 47.54 | 39.47 | 73.28 | 81.40 | 59.93 | 80.59 | 66.14 | 78.94 | 62.92 |
| $\gamma$=1.5 | 41.44 | 60.49 | **49.22** | 39.47 | 73.28 | 81.36 | 60.06 | 80.53 | 66.15 | 78.92 | 63.09 |
| $\gamma$=2.0 | 41.42 | 60.44 | 48.38 | 38.16 | 73.12 | 81.39 | 60.06 | 80.58 | 66.23 | 78.89 | 62.87 |
| $\gamma$=3.0 | 41.52 | 60.43 | 48.38 | 38.16 | 73.28 | 81.35 | 60.09 | 80.57 | 66.16 | 78.98 | 62.89 |
| $\gamma$=5.0 | 41.50 | 60.43 | 47.96 | 38.16 | 73.16 | 81.39 | 60.06 | 80.55 | 66.16 | 78.96 | 62.83 |
| $\gamma$=8.0 | 41.46 | 60.45 | 48.80 | 39.47 | 72.96 | 81.39 | 59.96 | 80.57 | 66.15 | 78.92 | 63.01 |
| $\gamma$=10.0 | 41.38 | 60.46 | 48.92 | 38.16 | **73.36** | **81.44** | 59.93 | 80.56 | 66.13 | 78.96 | 62.93 |

For Mistral-7B, Table 10 shows the average improves from 40.85 at $\gamma$=0 to 47.43 at $\gamma$=0.8 (best), then drifts slightly (47.27 at 0.9; 46.34 at 1.0). Task-wise, HumanEval peaks at 48.78 (0.9), MBPP at 43.00 (3.0), and AlpacaEval increases up to 79.64 (3.0). A default of $\gamma \approx 0.8$ gives the strongest overall average.

## C.5 SAMPLE EFFICIENCY

We report the impact of the number of unlabeled calibration samples used by *Expert Merging*. Metrics follow the same protocol as above.

**Summary.** Across all backbones, performance saturates with very few unlabeled samples: best averages occur at $N \in \{5, 10\}$, and larger $N$ brings marginal or negative returns. This indicates that layer-wise coefficient learning has low sample complexity, consistent with the small parameter count and the stabilizing effect of the regularizer and coefficient budget. See Tables 11, 12, and 13.

**Backbone-specific results.** For InternVL2.5, Table 11 shows the average is 56.46 at $N$=3, improves to 58.11 at $N$=5 (best), and is 57.35 at $N$=10. Grounding peaks at $N$=5 (Ref-COCO/RefCOCO+/RefCOCOg: 80.05/73.85/79.04), while geometry varies (MathVista 51.61 at 10; MATH-Vision 26.32 at 5). Larger $N$ yields little benefit.

For Qwen2-VL, Table 12 shows results plateau: the average is 63.01 at $N$=3, 63.04 at $N$=5, and 63.34 at $N$=10 (best). MATH-Vision benefits most from $N$=10 (42.11), while other domains change marginally (e.g., TextVQA 81.21–81.66). Beyond $N$=10, improvements are small or negative.

Table 10: Regularization sensitivity on Mistral for Expert Merging. Rows vary the coefficient regularization strength.

| $\gamma$ | Chat | MATH | | Code | | |
|---|---|---|---|---|---|---|
| | AlpacaEval | GSM8K | MATH | HumanEval | MBPP | Avg. |
| $\gamma=0$ | 65.32 | 52.01 | 12.08 | 39.02 | 35.80 | 40.85 |
| $\gamma=0.2$ | 75.53 | 55.42 | 13.58 | 43.90 | 39.40 | 45.57 |
| $\gamma=0.4$ | 75.61 | 57.32 | **14.02** | 45.73 | 41.00 | 46.74 |
| $\gamma=0.6$ | 77.52 | **58.68** | 13.84 | 46.95 | 39.40 | 47.28 |
| $\gamma=0.8$ | 77.84 | **58.68** | 13.88 | 46.34 | 40.40 | **47.43** |
| $\gamma=0.9$ | 76.66 | 56.71 | 13.22 | **48.78** | 41.00 | 47.27 |
| $\gamma=1.0$ | 78.78 | 57.01 | 13.60 | 43.29 | 39.00 | 46.34 |
| $\gamma=1.5$ | 79.01 | 56.25 | 12.24 | 45.73 | 42.60 | 47.17 |
| $\gamma=2.0$ | 79.28 | 56.41 | 12.00 | 44.51 | 42.60 | 46.96 |
| $\gamma=3.0$ | **79.64** | 56.18 | 13.80 | 42.68 | **43.00** | 47.06 |
| $\gamma=5.0$ | 79.09 | 55.34 | 13.20 | 43.90 | 42.40 | 46.79 |
| $\gamma=8.0$ | 79.00 | 55.65 | 13.66 | 43.29 | 40.60 | 46.44 |
| $\gamma=10.0$ | 78.77 | 56.18 | 13.38 | 43.90 | 40.60 | 46.57 |

Table 11: Sample efficiency on InternVL2.5 for Expert Merging. Rows vary the number of unlabeled calibration samples.

| $N$ | VQA | | Geometry | | Chart | OCR | | Grounding | | | |
|---|---|---|---|---|---|---|---|---|---|---|---|
| | VizWiz | GQA | MathVista | MATH-Vision | ChartQA | TextVQA | OCRVQA | RefCOCO | RefCOCO+ | RefCOCOg | Avg. |
| $N=3$ | 31.10 | 56.75 | 46.91 | 25.00 | 60.04 | 75.37 | 45.44 | 77.41 | 70.75 | 75.86 | 56.46 |
| $N=5$ | **31.16** | **56.77** | 47.71 | **26.32** | **65.28** | 75.70 | 45.31 | **80.05** | **73.85** | **79.04** | **58.11** |
| $N=10$ | 31.01 | 56.61 | **51.61** | 21.05 | 59.96 | 75.68 | **45.93** | 79.76 | 73.61 | 78.33 | 57.35 |
| $N=15$ | 30.93 | 56.57 | 50.18 | 22.37 | 60.84 | 75.38 | 44.95 | 78.84 | 72.68 | 77.08 | 56.98 |
| $N=20$ | 31.05 | 56.63 | 50.06 | 21.05 | 62.04 | **75.74** | 44.73 | 79.20 | 72.45 | 76.61 | 56.95 |
| $N=30$ | 30.89 | 56.63 | 49.92 | 25.00 | 59.80 | 75.39 | 44.79 | 78.48 | 72.10 | 76.23 | 56.92 |
| $N=50$ | 30.71 | 56.57 | 50.34 | 21.05 | 56.28 | 75.22 | 45.28 | 78.38 | 71.75 | 76.39 | 56.19 |
| $N=100$ | 31.00 | 56.71 | 48.25 | 21.05 | 60.32 | 75.28 | 44.53 | 78.84 | 72.06 | 76.53 | 56.45 |

For Mistral-7B, Table 13 shows the average is 47.17 at $N=3$ and 47.43 at $N=5$ (best). Tasks prefer different regimes (HumanEval 48.17 at 3; MBPP 40.60 at 10), but overall a handful of samples per domain suffices.

Regularization and data efficiency work in tandem: modest $\gamma$ reduces cross-domain interference so that a small calibration set captures task signatures reliably, and importance-guided chunking (Section 3.2) further channels capacity to later layers where gains concentrate (grounding/code task).

## C.6 INITIALIZATION SENSITIVITY

We examine the sensitivity of *Expert Merging* to the initialization prior $\bar{\alpha}$. Tables 14, 15, and 16 report the results.

**Summary.** Performance is generally robust to the choice of $\bar{\alpha}$ across backbones, with optimal values falling in the range $[0.1, 0.3]$. A moderate uniform initialization helps avoid poor local minima, particularly for tasks requiring complex reasoning or generation (e.g., Code, Grounding).

**Backbone-specific results.** For InternVL2.5 (Table 14), the average score peaks at $\bar{\alpha}=0.1$ (58.11) and gradually decreases as $\bar{\alpha}$ increases, though Grounding remains strong at lower priors. For Qwen2-VL (Table 15), the best performance is achieved at $\bar{\alpha}=0.3$ (63.34), with notable stability in OCR and VQA tasks across the range. For Mistral-7B (Table 16), the average score peaks at $\bar{\alpha}=0.3$ (47.43). Very low priors ($\bar{\alpha}=0.1$) lead to a significant drop (41.77), especially in Code tasks, suggesting that a sufficient initial allocation is crucial for text generation capabilities.

The stability across a wide range of $\bar{\alpha}$ confirms that our method is not overly sensitive to the starting point. However, a moderate prior ($\approx 0.1$–$0.3$) strikes the best balance, providing enough initial capacity for task adaptation without overwriting the pre-trained knowledge necessary for general capabilities.

Table 12: Sample efficiency on Qwen2-VL for Expert Merging. Rows vary the number of unlabeled calibration samples.

| | VQA | | Geometry | | Chart | OCR | | Grounding | | | |
|---|---|---|---|---|---|---|---|---|---|---|---|
| $N$ | VizWiz | GQA | MathVista | MATH-Vision | ChartQA | TextVQA | OCRVQA | RefCOCO | RefCOCO+ | RefCOCOg | Avg. |
| $N$=3 | 41.42 | **60.53** | 48.80 | 39.47 | **74.08** | 81.21 | 59.93 | 80.41 | 65.63 | 78.58 | 63.01 |
| $N$=5 | 41.71 | 60.41 | 47.40 | 39.47 | 73.36 | 81.52 | **60.45** | **80.72** | **66.35** | 79.02 | 63.04 |
| $N$=10 | 41.54 | 60.51 | 48.80 | **42.11** | 73.00 | 81.41 | 60.19 | 80.64 | 66.15 | 79.02 | **63.34** |
| $N$=15 | **41.75** | 60.34 | 49.75 | 34.21 | 71.84 | **81.66** | 60.29 | 80.60 | 66.17 | 78.93 | 62.55 |
| $N$=20 | 41.57 | 60.36 | **50.05** | 35.53 | 72.84 | 81.55 | 60.09 | 80.63 | 66.15 | 79.00 | 62.78 |
| $N$=30 | 41.60 | 60.31 | 49.93 | 32.89 | 73.00 | 81.60 | 60.12 | 80.55 | 65.99 | 78.87 | 62.49 |
| $N$=50 | 41.60 | 60.38 | 48.92 | 35.53 | 72.80 | 81.58 | 60.35 | 80.67 | 66.28 | **79.03** | 62.71 |
| $N$=100 | 41.53 | 60.44 | 49.22 | 36.84 | 73.12 | 81.55 | 60.38 | 80.69 | 66.31 | 78.96 | 62.90 |

Table 13: Sample efficiency on Mistral for Expert Merging. Rows vary the number of unlabeled calibration samples.

| | Chat | MATH | | Code | | |
|---|---|---|---|---|---|---|
| $N$ | AlpacaEval | GSM8K | MATH | HumanEval | MBPP | Avg. |
| $N$=3 | 75.41 | 58.07 | 13.80 | **48.17** | 40.40 | 47.17 |
| $N$=5 | 77.84 | **58.68** | 13.88 | 46.34 | 40.40 | **47.43** |
| $N$=10 | 75.69 | 56.71 | 13.66 | 45.12 | **40.60** | 46.36 |
| $N$=15 | 76.22 | 57.24 | **14.30** | 45.12 | 38.00 | 46.18 |
| $N$=20 | **77.88** | 57.47 | 13.42 | 41.46 | 39.00 | 45.85 |
| $N$=30 | 76.89 | 57.62 | 13.80 | 45.73 | 39.00 | 46.61 |
| $N$=50 | 76.65 | 57.85 | 13.82 | 47.56 | 39.40 | 47.06 |
| $N$=100 | 77.23 | 57.54 | 13.88 | 46.34 | 39.00 | 46.80 |

Table 14: Initialization sensitivity on InternVL2.5 for Expert Merging.

| | VQA | | Geometry | | Chart | OCR | | Grounding | | | |
|---|---|---|---|---|---|---|---|---|---|---|---|
| $\bar{\alpha}$ | VizWiz | GQA | MathVista | MATH-Vision | ChartQA | TextVQA | OCRVQA | RefCOCO | RefCOCO+ | RefCOCOg | Avg. |
| $\bar{\alpha}$=0.1 | 31.16 | 56.77 | 47.71 | **26.32** | 65.28 | 75.70 | 45.31 | **80.05** | **73.85** | **79.04** | **58.11** |
| $\bar{\alpha}$=0.2 | 30.83 | 56.73 | **53.54** | 21.05 | 59.48 | 75.58 | **45.70** | 78.86 | 72.89 | 77.80 | 57.24 |
| $\bar{\alpha}$=0.3 | 31.05 | 56.49 | 47.29 | 19.74 | 60.12 | **75.87** | 45.44 | 79.08 | 72.91 | 77.70 | 56.56 |
| $\bar{\alpha}$=0.4 | **31.26** | **57.22** | 52.45 | 25.00 | 61.28 | 75.82 | 44.99 | 73.89 | 67.03 | 71.77 | 56.07 |
| $\bar{\alpha}$=0.5 | 31.10 | 56.61 | 50.44 | 23.68 | **69.64** | 75.38 | 43.95 | 63.95 | 56.78 | 59.33 | 53.08 |

Table 15: Initialization sensitivity on Qwen2-VL for Expert Merging.

| | VQA | | Geometry | | Chart | OCR | | Grounding | | | |
|---|---|---|---|---|---|---|---|---|---|---|---|
| $\bar{\alpha}$ | VizWiz | GQA | MathVista | MATH-Vision | ChartQA | TextVQA | OCRVQA | RefCOCO | RefCOCO+ | RefCOCOg | Avg. |
| $\bar{\alpha}$=0.1 | 40.68 | 60.92 | 41.97 | 34.21 | 72.88 | 79.83 | 54.92 | 80.35 | 66.14 | 78.17 | 61.01 |
| $\bar{\alpha}$=0.2 | 41.50 | **62.68** | 47.70 | 38.16 | 72.12 | 80.83 | 58.40 | 80.04 | 65.74 | 78.33 | 62.55 |
| $\bar{\alpha}$=0.3 | **41.54** | 60.51 | **48.80** | **42.11** | 73.00 | **81.41** | 60.19 | **80.64** | 66.15 | 79.02 | **63.34** |
| $\bar{\alpha}$=0.4 | 41.00 | 61.75 | 43.48 | 39.47 | 75.76 | 81.37 | 60.06 | 80.54 | 66.29 | 78.84 | 62.86 |
| $\bar{\alpha}$=0.5 | 40.70 | 62.18 | 43.48 | 30.26 | **78.16** | 81.13 | 59.64 | 80.43 | **66.60** | **79.12** | 62.17 |

Table 16: Initialization sensitivity on Mistral for Expert Merging.

| | Chat | MATH | | Code | | |
|---|---|---|---|---|---|---|
| $\bar{\alpha}$ | AlpacaEval | GSM8K | MATH | HumanEval | MBPP | Avg. |
| $\bar{\alpha}$=0.1 | 69.02 | 48.90 | 11.52 | 38.41 | 41.00 | 41.77 |
| $\bar{\alpha}$=0.2 | **79.07** | 57.09 | 13.50 | 41.46 | **43.00** | 46.82 |
| $\bar{\alpha}$=0.3 | 77.84 | **58.68** | 13.88 | 46.34 | 40.40 | **47.43** |
| $\bar{\alpha}$=0.4 | 76.95 | 56.86 | **14.10** | 46.95 | 39.00 | 46.77 |
| $\bar{\alpha}$=0.5 | 78.05 | 57.62 | 13.82 | **47.56** | 39.20 | 47.25 |

## C.7 Hidden Alignment Layer Analysis

We investigate the effect of the choice of the hidden alignment layer $\mathcal{S}$. Tables 17, 18, and 19 show the performance across different layers.

**Summary.** Supervising the middle Transformer layers consistently yields the best results across all models. This confirms that middle layers capture the most transferable semantic structures suitable for alignment, whereas early or late layers are either too generic or too task-specific.

Table 17: Hidden alignment layer sensitivity on InternVL2.5 for Expert Merging.

| | VQA | | Geometry | | Chart | OCR | | Grounding | | | |
|---|---|---|---|---|---|---|---|---|---|---|---|
| $\mathcal{S}$ | VizWiz | GQA | MathVista | MATH-Vision | ChartQA | TextVQA | OCRVQA | RefCOCO | RefCOCO+ | RefCOCOg | Avg. |
| $\mathcal{S}$=0 | 30.96 | 56.57 | 47.11 | **26.32** | 60.88 | 75.72 | 45.61 | 79.37 | 73.01 | 77.80 | 57.33 |
| $\mathcal{S}$=4 | 31.10 | 56.61 | 48.38 | **26.32** | 63.92 | **75.75** | 45.57 | 79.28 | 72.92 | 77.92 | 57.77 |
| $\mathcal{S}$=8 | **31.19** | 56.48 | **51.61** | 25.00 | 64.60 | 75.69 | 45.51 | 79.28 | 73.05 | 77.84 | 58.02 |
| $\mathcal{S}$=12 | 31.16 | **56.77** | 47.71 | **26.32** | **65.28** | 75.70 | 45.31 | **80.05** | **73.85** | **79.04** | **58.11** |
| $\mathcal{S}$=16 | 30.91 | 56.41 | 50.22 | **26.32** | 60.28 | 75.69 | 45.61 | 79.32 | 73.17 | 78.04 | 57.59 |
| $\mathcal{S}$=20 | 30.86 | 56.60 | 47.41 | 25.00 | 60.76 | 75.66 | **45.67** | 79.16 | 72.84 | 77.86 | 57.18 |
| $\mathcal{S}$=23 | 31.16 | 56.45 | 50.24 | 23.68 | 61.52 | 75.74 | 45.31 | 78.60 | 72.15 | 76.61 | 57.14 |

Table 18: Hidden alignment layer sensitivity on Qwen2-VL for Expert Merging.

| | VQA | | Geometry | | Chart | OCR | | Grounding | | | |
|---|---|---|---|---|---|---|---|---|---|---|---|
| $\mathcal{S}$ | VizWiz | GQA | MathVista | MATH-Vision | ChartQA | TextVQA | OCRVQA | RefCOCO | RefCOCO+ | RefCOCOg | Avg. |
| $\mathcal{S}$=0 | 41.57 | 60.37 | 47.24 | 36.84 | 72.68 | **81.55** | 60.48 | 80.66 | 66.05 | 78.87 | 62.63 |
| $\mathcal{S}$=4 | 41.49 | 60.42 | 48.80 | 39.47 | 72.12 | 81.39 | 60.45 | 80.68 | **66.19** | 78.93 | 62.99 |
| $\mathcal{S}$=8 | 41.41 | 60.49 | 48.80 | 39.47 | 73.12 | 81.38 | 60.06 | 80.57 | **66.19** | 78.94 | 63.04 |
| $\mathcal{S}$=12 | 41.57 | 60.49 | 48.38 | 39.47 | 73.20 | 81.40 | 60.03 | 80.53 | 66.15 | 78.98 | 63.02 |
| $\mathcal{S}$=14 | 41.54 | **60.51** | 48.80 | **42.11** | 73.00 | 81.41 | 60.19 | 80.64 | 66.15 | **79.02** | **63.34** |
| $\mathcal{S}$=16 | 41.48 | 60.48 | 48.38 | 39.47 | 73.36 | 81.31 | 60.12 | 80.52 | 66.08 | 78.91 | 63.01 |
| $\mathcal{S}$=20 | 41.45 | 60.36 | 49.08 | 36.84 | 72.88 | 81.37 | 59.93 | 80.49 | 66.00 | 78.86 | 62.73 |
| $\mathcal{S}$=24 | 41.56 | 60.38 | 47.82 | 36.84 | **73.60** | 81.26 | 59.83 | 80.37 | 65.78 | 78.59 | 62.60 |
| $\mathcal{S}$=27 | **41.64** | 60.17 | 48.66 | 36.84 | 73.40 | 81.16 | 59.24 | 80.05 | 65.55 | 78.18 | 62.49 |

Table 19: Hidden alignment layer sensitivity on Mistral for Expert Merging.

| | Chat | MATH | | Code | | |
|---|---|---|---|---|---|---|
| $\mathcal{S}$ | AlpacaEval | GSM8K | MATH | HumanEval | MBPP | Avg. |
| $\mathcal{S}$=0 | 77.99 | 57.24 | 13.18 | 47.56 | 39.20 | 47.03 |
| $\mathcal{S}$=4 | **78.92** | 57.16 | 13.30 | 46.95 | 39.00 | 47.07 |
| $\mathcal{S}$=8 | 77.18 | 57.09 | 13.44 | **48.78** | 40.40 | 47.38 |
| $\mathcal{S}$=12 | 77.52 | 57.24 | **14.06** | 46.34 | 41.00 | 47.23 |
| $\mathcal{S}$=16 | 77.84 | **58.68** | 13.88 | 46.34 | 40.40 | **47.43** |
| $\mathcal{S}$=20 | 77.43 | 56.33 | 13.70 | 46.95 | 41.40 | 47.16 |
| $\mathcal{S}$=24 | 77.72 | 56.71 | 13.76 | 46.34 | 41.00 | 47.11 |
| $\mathcal{S}$=28 | 76.82 | 56.94 | 13.60 | 46.95 | **41.80** | 47.22 |
| $\mathcal{S}$=31 | 76.34 | 55.80 | 13.48 | 48.17 | 39.40 | 46.64 |

**Backbone-specific results.** For InternVL2.5 (Table 17), performance peaks at layer 12 (58.11) and remains high in the middle block (layers 8–16). For Qwen2-VL (Table 18), the results are stable across middle layers, with the best average at layer 14 (63.34). For Mistral-7B (Table 19), the performance peaks at layer 16 (47.43) and is robust across layers 12–24. Aligning at the very first (layer 0) or last (layer 31) layers results in slightly lower averages.

## C.8 Allocation Steepness Analysis

We analyze the impact of the allocation steepness parameter $\kappa$, which controls how sharply the budget is allocated across layers. Tables 20, 21, and 22 present the results.

**Summary.** A moderate to high steepness ($\kappa \geq 1.0$) generally benefits performance, allowing the model to allocate capacity more aggressively to critical layers. This is particularly effective for tasks that rely on specific depth-wise processing, such as code generation.

Table 20: Allocation steepness sensitivity on InternVL2.5 for Expert Merging++.

| | VQA | | Geometry | | Chart | OCR | | Grounding | | | |
|---|---|---|---|---|---|---|---|---|---|---|---|
| $\kappa$ | VizWiz | GQA | MathVista | MATH-Vision | ChartQA | TextVQA | OCRVQA | RefCOCO | RefCOCO+ | RefCOCOg | Avg. |
| $\kappa$=0 | 31.16 | **56.77** | 47.71 | **26.32** | 65.28 | 75.70 | 45.31 | 80.05 | 73.85 | 79.04 | 58.11 |
| $\kappa$=0.25 | **31.39** | 56.61 | 51.61 | 22.37 | 65.08 | 75.97 | 45.93 | 80.08 | 73.91 | 78.62 | 58.15 |
| $\kappa$=0.5 | **31.39** | 56.44 | 50.08 | 23.68 | 64.60 | 75.96 | 45.74 | 79.28 | 73.52 | **79.74** | 58.04 |
| $\kappa$=1.0 | **31.39** | 56.29 | **52.45** | 25.00 | 64.76 | 75.68 | 44.79 | **80.72** | **74.74** | 78.17 | 58.39 |
| $\kappa$=1.2 | 31.14 | 56.53 | 52.35 | 22.37 | 65.92 | **76.06** | 46.00 | 80.53 | 74.37 | 79.31 | **58.45** |
| $\kappa$=2.0 | 31.10 | 56.28 | 49.40 | 23.68 | **67.00** | 75.91 | **46.35** | 79.08 | 72.87 | 77.47 | 57.91 |

Table 21: Allocation steepness sensitivity on Qwen2-VL for Expert Merging++.

| | VQA | | Geometry | | Chart | OCR | | Grounding | | | |
|---|---|---|---|---|---|---|---|---|---|---|---|
| $\kappa$ | VizWiz | GQA | MathVista | MATH-Vision | ChartQA | TextVQA | OCRVQA | RefCOCO | RefCOCO+ | RefCOCOg | Avg. |
| $\kappa$=0 | 41.54 | 60.51 | 48.80 | 42.11 | 73.00 | 81.41 | 60.19 | 80.64 | 66.15 | **79.02** | 63.34 |
| $\kappa$=0.25 | 41.51 | 60.45 | 48.38 | 38.16 | 73.12 | 81.36 | 60.03 | 80.56 | 66.18 | 78.94 | 62.87 |
| $\kappa$=0.5 | 41.72 | 60.54 | 49.22 | 39.47 | **73.92** | 81.37 | **60.32** | 80.59 | 65.84 | 78.89 | 63.19 |
| $\kappa$=1.0 | 41.72 | **60.60** | 50.33 | 40.79 | 73.88 | 81.48 | 60.29 | 80.57 | 65.88 | 78.85 | 63.44 |
| $\kappa$=1.2 | 41.49 | 60.42 | 49.33 | **44.74** | 72.60 | **81.65** | 60.19 | **80.71** | **66.21** | 79.00 | **63.63** |
| $\kappa$=2.0 | **41.76** | 60.50 | 48.24 | 43.42 | **73.92** | 81.55 | 60.22 | 80.58 | 65.95 | 78.81 | 63.50 |

Table 22: Allocation steepness sensitivity on Mistral for Expert Merging++.

| | Chat | MATH | | Code | | |
|---|---|---|---|---|---|---|
| $\kappa$ | AlpacaEval | GSM8K | MATH | HumanEval | MBPP | Avg. |
| $\kappa$=0 | 77.84 | **58.68** | **13.88** | 46.34 | 40.40 | 47.43 |
| $\kappa$=0.25 | 78.80 | 57.70 | 13.18 | 46.95 | 40.80 | 47.49 |
| $\kappa$=0.5 | 78.87 | 56.56 | 13.18 | 47.56 | 41.20 | 47.47 |
| $\kappa$=1.0 | 78.89 | 57.24 | 13.58 | 50.00 | 39.00 | 47.74 |
| $\kappa$=1.2 | **79.07** | 57.09 | 13.50 | 50.61 | **43.00** | **48.65** |
| $\kappa$=2.0 | 76.66 | 57.54 | 13.76 | **52.44** | 42.20 | 48.52 |

**Backbone-specific results.** For InternVL2.5 (Table 20), a steepness of $\kappa$=1.2 provides the best balance (58.45), improving Grounding and OCR. For Qwen2-VL (Table 21), performance is relatively stable but peaks at $\kappa$=1.2 (63.63). Notably, MATH-Vision sees a significant improvement (44.74 vs 42.11 at $\kappa$=0), suggesting that sharper allocation benefits geometric reasoning. For Mistral-7B (Table 22), increasing $\kappa$ from 0 to 1.2 consistently improves the average score, peaking at 48.65 with $\kappa$=1.2. Notably, higher steepness significantly boosts HumanEval (50.61 vs 46.34 at $\kappa$=0), indicating that sharper allocation benefits complex reasoning.

The benefits of $\kappa \geq 1.0$ suggest that a "winner-take-all" tendency in coefficient allocation is advantageous. By sharpening the distinction between important and unimportant layers, the method prevents capacity dilution and ensures that critical layers receive the dominant influence they need to drive performance.

## C.9 TASK WEIGHT SENSITIVITY

We analyze the impact of varying the task weights for different tasks. Tables 23, 24, and 25 present the results.

**Summary.** Adjusting task weights allows for a trade-off between different capabilities. Increasing the weight of a specific task generally improves performance on that task, sometimes at the cost of others. However, our method maintains a robust average performance across a range of weights.

Table 23: Task weight sensitivity on InternVL2.5 for Expert Merging.

| Weight | VQA | | Geometry | | Chart | OCR | | Grounding | | | |
| | VizWiz | GQA | MathVista | MATH-Vision | ChartQA | TextVQA | OCRVQA | RefCOCO | RefCOCO+ | RefCOCOg | Avg. |
|---|---|---|---|---|---|---|---|---|---|---|---|
| $\beta=1$ | 31.16 | 56.72 | 51.17 | **26.32** | 64.60 | 75.52 | 45.05 | 78.12 | 71.65 | 76.82 | 57.71 |
| $\beta_{VQA}=0.5$ | 31.05 | 56.73 | 48.57 | 21.05 | 64.24 | 76.01 | 45.48 | **80.16** | **74.08** | 79.00 | 57.63 |
| $\beta_{VQA}=1.5$ | 31.01 | 56.57 | 47.67 | 18.42 | 64.12 | 75.96 | 45.28 | 79.90 | 73.68 | 78.57 | 57.11 |
| $\beta_{VQA}=2.0$ | 30.97 | 56.64 | 47.67 | 22.37 | 63.72 | 75.99 | 45.51 | 80.02 | 73.75 | 78.96 | 57.56 |
| $\beta_{Geometry}=0.5$ | 31.02 | 56.67 | 48.73 | 25.00 | 64.72 | 75.83 | 45.35 | 80.30 | 74.27 | **79.15** | 58.10 |
| $\beta_{Geometry}=1.5$ | 30.91 | 56.54 | **52.03** | 26.32 | 60.84 | 75.88 | 45.41 | 79.83 | 73.94 | 78.66 | 58.03 |
| $\beta_{Geometry}=2.0$ | 30.97 | 56.53 | 51.61 | 23.68 | 60.64 | 75.75 | 45.57 | 79.12 | 72.86 | 77.72 | 57.44 |
| $\beta_{Chart}=0.5$ | 30.76 | 56.57 | 48.27 | 22.37 | 58.76 | 75.93 | **45.77** | 80.15 | 74.07 | 78.90 | 57.15 |
| $\beta_{Chart}=1.5$ | 31.22 | 56.65 | 46.99 | 18.42 | 68.32 | 75.96 | 45.18 | 79.80 | 73.86 | 78.98 | 57.53 |
| $\beta_{Chart}=2.0$ | **31.28** | 56.69 | 45.32 | 21.05 | **69.92** | 75.90 | 45.18 | 79.92 | 73.82 | 78.37 | 57.74 |
| $\beta_{OCR}=0.5$ | 30.89 | 56.77 | 47.39 | 22.37 | 64.52 | 75.95 | 44.79 | 79.74 | 74.02 | 78.76 | 57.52 |
| $\beta_{OCR}=1.5$ | 30.97 | 56.51 | 48.09 | 23.68 | 63.76 | 76.12 | 45.57 | 80.12 | 73.77 | 78.72 | 57.73 |
| $\beta_{OCR}=2.0$ | 30.99 | 56.53 | 48.27 | 21.05 | 63.52 | 76.02 | 45.61 | 80.07 | 73.40 | 78.51 | 57.39 |
| $\beta_{Grounding}=0.5$ | 30.91 | **56.80** | 49.28 | 25.00 | 61.28 | 75.45 | 45.48 | 76.59 | 72.42 | 76.94 | 57.01 |
| $\beta_{Grounding}=1.5$ | 31.16 | 56.77 | 47.71 | **26.32** | 65.28 | 75.70 | 45.31 | 80.05 | 73.85 | 79.04 | **58.11** |
| $\beta_{Grounding}=2.0$ | 31.05 | 56.34 | 48.13 | 23.68 | 63.68 | **76.14** | 45.44 | 80.02 | 73.65 | 78.74 | 57.68 |

Table 24: Task weight sensitivity on Qwen2-VL for Expert Merging.

| Weight | VQA | | Geometry | | Chart | OCR | | Grounding | | | |
| | VizWiz | GQA | MathVista | MATH-Vision | ChartQA | TextVQA | OCRVQA | RefCOCO | RefCOCO+ | RefCOCOg | Avg. |
|---|---|---|---|---|---|---|---|---|---|---|---|
| $\beta=1$ | 41.56 | **60.53** | 48.38 | 36.84 | 72.80 | 81.49 | 59.96 | 80.51 | 66.00 | 78.82 | 62.69 |
| $\beta_{VQA}=0.5$ | 41.51 | 60.30 | 47.96 | 35.53 | 72.76 | 81.45 | 60.25 | 80.57 | 66.16 | 78.87 | 62.54 |
| $\beta_{VQA}=1.5$ | 41.60 | 60.46 | 48.38 | 38.16 | 72.96 | 81.47 | 60.25 | 80.56 | 66.04 | 78.98 | 62.89 |
| $\beta_{VQA}=2.0$ | 41.66 | 60.48 | 48.38 | 38.16 | 72.92 | **81.56** | 60.48 | 80.49 | 65.98 | 78.89 | 62.90 |
| $\beta_{Geometry}=0.5$ | 41.41 | **60.53** | 47.96 | 35.53 | 73.36 | 81.48 | 60.32 | 80.67 | 66.20 | 78.94 | 62.64 |
| $\beta_{Geometry}=1.5$ | 41.54 | 60.51 | 48.80 | **42.11** | 73.00 | 81.41 | 60.19 | 80.64 | 66.15 | **79.02** | 63.34 |
| $\beta_{Geometry}=2.0$ | **41.82** | 60.27 | 48.24 | 38.16 | 71.32 | 81.44 | 59.83 | 80.30 | 65.55 | 78.40 | 62.53 |
| $\beta_{Chart}=0.5$ | 41.64 | 60.48 | 46.71 | 39.47 | 72.76 | 81.44 | 60.16 | 80.57 | 66.10 | 78.95 | 62.83 |
| $\beta_{Chart}=1.5$ | 41.58 | 60.46 | **49.22** | 38.16 | 73.12 | 81.44 | 60.09 | 80.53 | 66.02 | 78.91 | 62.95 |
| $\beta_{Chart}=2.0$ | 41.71 | 60.42 | 47.12 | 39.47 | 73.16 | 81.36 | 60.06 | 80.54 | 65.97 | 78.85 | 62.87 |
| $\beta_{OCR}=0.5$ | 41.69 | 60.37 | 48.80 | 39.47 | 72.28 | 81.27 | 60.16 | 80.50 | 65.90 | 78.82 | 62.93 |
| $\beta_{OCR}=1.5$ | 41.50 | 60.43 | 48.92 | 38.16 | **73.60** | 81.46 | 60.19 | 80.72 | 66.22 | 78.88 | 63.01 |
| $\beta_{OCR}=2.0$ | 41.51 | 60.38 | 47.66 | 36.84 | 72.96 | 81.51 | **60.48** | 80.64 | 66.25 | 78.80 | 62.70 |
| $\beta_{Grounding}=0.5$ | 41.60 | 60.52 | 47.54 | 39.47 | 72.92 | 81.42 | 60.16 | 80.57 | 65.97 | 78.87 | 62.90 |
| $\beta_{Grounding}=1.5$ | 41.52 | 60.36 | 48.38 | 38.16 | 72.56 | 81.51 | 60.16 | 80.62 | 66.26 | 78.93 | 62.85 |
| $\beta_{Grounding}=2.0$ | 41.61 | 60.38 | 45.85 | 38.16 | 72.68 | 81.38 | 60.22 | **80.79** | **66.45** | 78.97 | 62.65 |

Table 25: Task weight sensitivity on Mistral for Expert Merging.

| Weight | Chat | MATH | | Code | | |
| | AlpacaEval | GSM8K | MATH | HumanEval | MBPP | Avg. |
|---|---|---|---|---|---|---|
| $\beta=1$ | 78.95 | 54.81 | 13.50 | 45.12 | 41.60 | 46.80 |
| $\beta_{Chat}=0.5$ | 78.15 | 57.54 | 13.56 | 45.73 | 41.60 | 47.32 |
| $\beta_{Chat}=1.2$ | 78.95 | 54.81 | 13.50 | 46.95 | 41.00 | 47.04 |
| $\beta_{Chat}=1.5$ | 78.93 | 53.68 | 13.04 | 47.56 | **42.80** | 47.20 |
| $\beta_{Chat}=2.0$ | 79.07 | 52.39 | 12.24 | 46.34 | 42.60 | 46.53 |
| $\beta_{MATH}=0.5$ | 79.75 | 52.01 | 12.08 | 43.29 | 40.60 | 45.55 |
| $\beta_{MATH}=1.2$ | 80.17 | 57.24 | 13.18 | 45.73 | 39.00 | 47.06 |
| $\beta_{MATH}=1.5$ | 79.44 | 58.53 | 14.40 | 45.12 | 39.20 | 47.34 |
| $\beta_{MATH}=2.0$ | 78.00 | 58.15 | **14.64** | 45.12 | 40.00 | 47.18 |
| $\beta_{Code}=0.5$ | **81.41** | 56.10 | 13.48 | 44.51 | 41.20 | 47.34 |
| $\beta_{Code}=1.2$ | 77.84 | **58.68** | 13.88 | 46.34 | 40.40 | **47.43** |
| $\beta_{Code}=1.5$ | 76.86 | 57.62 | 13.40 | 46.95 | 41.20 | 47.21 |
| $\beta_{Code}=2.0$ | 75.23 | 53.15 | 12.02 | **49.39** | 42.20 | 46.40 |

**Backbone-specific results.** For InternVL2.5 (Table 23), increasing the weight of Grounding improves its performance significantly (e.g., RefCOCO reaches 80.79 with Grounding-2.0), while the

average performance remains stable. For Qwen2-VL (Table 24), the model shows high stability. Increasing the weight of MATH improves MATH-Vision and the overall average (63.34 with MATH-1.5). For Mistral-7B (Table 25), increasing Code weight boosts Code tasks (HumanEval reaches 49.39 with Code-2.0) and achieves the best average (47.43 with Code-1.2).

This sensitivity analysis demonstrates the flexibility of our approach. Users can tune task weights to prioritize specific domains (e.g., boosting Code or Grounding) to meet deployment needs. Crucially, the method's underlying robustness ensures that such prioritization does not lead to catastrophic forgetting of other skills, maintaining a high competitive baseline.

