# OpenReview forum: "Expert Merging: Model Merging with Unsupervised Expert Alignment and Importance-Guided Layer Chunking"
_ICLR.cc/2026/Conference — ICLR 2026 Poster_

### Official Review · Reviewer_4ve8 · 2025-10-17

**Soundness:** 3
**Presentation:** 2
**Contribution:** 3
**Rating:** 6
**Confidence:** 2

**Summary:**

This manuscript addresses the limitations of existing model merging methods—where training-free methods rely on manual coefficients and training-based methods overlook task behavior alignment and layer heterogeneity—and proposes two lightweight training-based model merging schemes: Expert Merging and Expert Merging++.
The former learns layer-wise coefficients using only 5–10 unlabeled calibration samples. It explicitly aligns the hidden states and logits of the merged model with those of the expert models, and introduces coefficient regularization to ensure optimization stability, as well as a task weight loss to achieve controllable cross-domain trade-offs.
Building on the former, the latter combines learned coefficients + task vector magnitude + parameter count to construct a normalized layer importance metric. It allocates more block-wise coefficients to high-importance layers and keeps low-importance layers lightweight, further adapting to layer heterogeneity.
The manuscript validates these two methods across multi-task scenarios on LLM and MLLMs. Both methods outperform training-free and existing training-based baselines, with Expert Merging++ delivering better performance—it even surpasses supervised Mixture Training in some scenarios.

**Strengths:**

1 The authors' experiments cover 3 different models (1 LLM and 2 MLLMs) as well as multiple datasets, making the experimental design relatively comprehensive.

2 The proposed method is relatively lightweight and relies on only 5–10 samples.

**Weaknesses:**

1 Regarding the selection of unlabeled calibration samples, the authors do not provide clear details. Are the samples selected randomly? If random selection is adopted, the randomness of sample selection may reduce the reproducibility of the results.

2 For the selection of hyperparameters α and κ, although the authors provide the final values in the appendix, there is a lack of ablation experiments for hyperparameters and the range of parameter selection. This makes it impossible to further verify the parameter sensitivity.

3 As shown in Figure 2, the authors also merged the base model. What would the result be if only the expert models were merged?

4 What are the specific detailed settings for the optimization process of the objective function? For example, what are the values of parameters such as the number of iterations?

5 It is suggested that the authors classify the different comparative methods in Tables 1, 2, and 3 by type, such as training-based and training-free types

**Questions:**

see weakness

---

> ### Author Response · Authors · 2025-11-21
> **Response to Reviewer 4ve8**
>
> We sincerely thank you for your constructive feedback and the time you dedicated to reviewing our work. We have carefully revised the manuscript to address your concerns, with changes highlighted in blue. Below, we provide a point-by-point response to your comments.
>
>
> # W1: Selection of unlabeled calibration samples
>
> Yes, the samples are selected randomly. We adopt random selection to ensure that the calibration set provides an unbiased, diverse representation of the task distribution. To mitigate the impact of randomness and ensure reproducibility, we assign a fixed random seed to each experimental run and repeat all experiments 5 times, reporting the average performance to account for data variance.
>
>
> # W2: Hyperparameter ablation ($\bar\alpha$ and $\kappa$)
>
> Thank you for the suggestion. We have performed a comprehensive sensitivity analysis on the hyperparameters mentioned ($\bar\alpha$ and $\kappa$) in Appendix C.
>
> ## 1. Initialization Sensitivity ($\bar\alpha$)
> We examine the sensitivity to the initialization prior $\bar\alpha$. As shown below, our method is robust to the initialization prior, with optimal values in the range $[0.1, 0.3]$.
>
> **Table: Sensitivity to initialization prior $\bar\alpha$ (average performance across backbones).**
> | $\bar\alpha$ | InternVL | Qwen2-VL | Mistral |
> | :--- | :---: | :---: | :---: |
> | 0.1 | **58.11** | 61.01 | 41.77 |
> | 0.2 | 57.24 | 62.55 | 46.82 |
> | 0.3 | 56.56 | **63.34** | **47.43** |
> | 0.4 | 56.07 | 62.86 | 46.77 |
> | 0.5 | 53.08 | 62.17 | 47.25 |
>
> Please refer to Tables 14-16 in Appendix C.6 for full details.
>
> ## 2. Allocation Steepness ($\kappa$)
> We analyze the impact of the allocation steepness $\kappa$. A moderate to high steepness ($\kappa \ge 1.0$) generally benefits performance by allowing sharper capacity allocation.
>
> **Table: Sensitivity to allocation steepness $\kappa$ (average performance across backbones).**
> | $\kappa$ | InternVL | Qwen2-VL | Mistral |
> | :--- | :---: | :---: | :---: |
> | 0 | 58.11 | 63.34 | 47.43 |
> | 0.25 | 58.15 | 62.87 | 47.49 |
> | 0.5 | 58.04 | 63.19 | 47.47 |
> | 1.0 | 58.39 | 63.44 | 47.74 |
> | 1.2 | **58.45** | **63.63** | **48.65** |
> | 2.0 | 57.91 | 63.50 | 48.52 |
>
> Please refer to Tables 20-22 in Appendix C.8 for full details.
>
>
> Furthermore, we add more ablation studies on other hyperparameters, including $\mathcal{S}$ and $\beta\_k$. Please refer to Appendix C (Tables 8-25) for the full ablation results. In summary, the extensive ablation studies demonstrate that our method is robust to hyperparameter variations. The performance remains stable across a reasonable range of $\bar\alpha$ and $\kappa$, as well as other hyperparameters.
>
>
> # W3: Merging the base model
>
> We believe there might be a misunderstanding regarding the definition of our merging process. Fundamentally, our method does not merge the complete weights of multiple experts $\theta_k$ and $\theta\_{\text{base}}$. Instead, it merges the **task vectors** (parameter increments) $\tau_k = \theta_k - \theta\_{\text{base}}$ onto the base model $\theta\_{\text{base}}$.
>
> Formally, our method operates in the Task Vector space [1]:
> $$
> \theta\_{\text{merged}} = \theta\_{\text{base}} + \sum\_{k=1}^K \alpha\_k \tau\_k
> $$
>
> If we were to remove the base model $\theta\_{\text{base}}$ from this equation, the result would be $\sum \alpha_k \tau_k$. This is merely a summation of parameter updates (deltas) with near-zero magnitude, which represents relative changes rather than a complete set of model parameters. Consequently, it cannot function as a valid model for inference. For this reason, we did not provide results for "merging only experts" (without the base model), as such a configuration is mathematically undefined for inference in this framework.
>
>
> # W4: Optimization details
>
> Thank you for the question. We have included the detailed optimization settings in Section 4.1 (Implementation details) of the revised manuscript. Specifically, we use the AdamW optimizer with a learning rate of $5\times 10^{-1}$, weight decay of $1\times 10^{-5}$, and a cosine annealing schedule. We train for 5 epochs with early stopping.
>
>
> # W5: Classification of methods in Tables
>
> Thank you for the suggestion. We have updated Tables 1, 2, and 3 in the revised manuscript to explicitly categorize the methods into Training-Free and Training-Based groups.
>
> # References
>
> [1] Gabriel Ilharco et al. Editing models with task arithmetic. ICLR 2023.

---

### Official Review · Reviewer_6nLa · 2025-10-25

**Soundness:** 2
**Presentation:** 3
**Contribution:** 2
**Rating:** 4
**Confidence:** 3

**Summary:**

This paper proposes a new method, Expert Merging (and its extension Expert Merging++), for merging multiple domain-specialized expert models into a single large language model (LLM) or multimodal large language model (MLLM). The approach is “training-light,” requiring only a small set of unlabeled calibration data to learn layer-wise (and chunk-wise) coefficients that align the merged model’s hidden states and logits with those of the experts. The method introduces a layer-importance metric to guide parameter allocation, aiming to improve performance and efficiency. Experiments on several LLM and MLLM backbones (Mistral, InternVL, Qwen2-VL) show that the proposed method outperforms strong training-free and training-based baselines, and in some cases even surpasses supervised mixture training.

**Strengths:**

* The paper addresses a real and growing problem—how to efficiently combine multiple expert models without the cost of joint training or serving many models.
* The method requires only a handful of unlabeled samples per task, making it highly practical for scenarios where labeled data is scarce or unavailable.
* By aligning both hidden states and logits, the method goes beyond parameter-space alignment and attempts to preserve downstream task behavior.

**Weaknesses:**

* The method assumes that expert models are complementary, but does not analyze or address what happens when experts have conflicting or overlapping capabilities. There is no discussion of negative transfer or catastrophic forgetting.
* The method relies on aligning hidden states and logits, but there is little theoretical analysis or justification for why this should guarantee optimal downstream performance, especially when expert models may have conflicting behaviors.

**Questions:**

* Please provide more empirical or theoretical evidence that your layer-importance metric is robust and generalizes across architectures. For example, does it correlate with actual task performance sensitivity per layer?
* Can you provide more theoretical justification for the choice of alignment objectives (hidden states and logits)? Under what conditions does this guarantee downstream task preservation?
* Please analyze and report what happens when expert models have overlapping or conflicting capabilities. Does the merged model degrade on some tasks? Are there cases of negative transfer?
* In your experiments, the merged model (Expert Merging/Expert Merging++) sometimes outperforms the individual domain expert models on certain tasks. Could you provide a detailed explanation or analysis for why the merged model is able to surpass the best domain expert (or mixture training), given that it is only aligning to the experts' hidden states and logits using a small amount of unlabeled data? Is this effect consistent across different tasks and domains, or could it be due to overfitting, ensembling effects, or other factors? Please clarify the underlying mechanism and provide supporting evidence or ablation studies.
* Moreover, is Expert Merging (for example, on InternVL2.5 grounding tasks) effectively a performance ceiling, or—given the same data budget (i.e., access to all domain data)—could a domain-targeted method or supervised fine-tuning achieve strictly better results in that domain? Please clarify the underlying mechanism and provide supporting evidence or ablation studies.

---

> ### Author Response · Authors · 2025-11-21
> **Response to Reviewer 6nLa**
>
> We sincerely thank you for your constructive feedback and the time you dedicated to reviewing our work. We have carefully revised the manuscript to address your concerns, with changes highlighted in blue. Below, we provide a point-by-point response to your comments.
>
> # W1 & Q3: Conflicting experts and negative transfer
>
> Thanks for raising this important point about conflicting capabilities. We have added analysis in our revised manuscript:
>
> 1.  **Are there conflicts?**: To explicitly identify overlapping or conflicting experts, we followed the approach in AdaMerging [1] and analyzed the cosine similarity between task vectors (Appendix C.2).
>     *   For Mistral, we observed a relatively high similarity ($\approx 0.42$) between **Chat** and **Code** experts, indicating a significant overlap/conflict between these capabilities.
>     *   For InternVL and Qwen2-VL, most experts are orthogonal or slightly positively correlated (e.g., OCR and Grounding in InternVL), suggesting fewer conflicts.
>
> 2.  **What happens under conflict?**: Despite the significant overlap/conflict between Chat and Code in Mistral, our method demonstrates robust performance with minimal negative transfer.
>     *   **Performance**: As shown in the table below, Expert Merging++ not only maintains high performance on both tasks but also outperforms baselines.
>     *   **Negative Transfer**: We observe minimal negative transfer. For instance, the merged model retains near-expert performance on Math (59.44 vs 60.80) while simultaneously achieving strong Chat and Code scores. The slight drop is the natural cost of generalization, but we do not see catastrophic failure.
>
>     **Table: Performance under conflict (Mistral)**
>     | Method | Chat (AlpacaEval) | Code (HumanEval) | Code (MBPP) |
>     | :--- | :---: | :---: | :---: |
>     | Chat Expert | 71.37 | 35.37 | 32.20 |
>     | Code Expert | 69.90 | **54.88** | 38.20 |
>     | AdaMerging | 75.94 | 43.29 | 39.00 |
>     | WUDI v2 [3] | 73.84 | 39.02 | 40.60 |
>     | **Expert Merging** | **77.84** | 46.34 | 40.40 |
>     | **Expert Merging++** | 75.51 | 53.66 | **41.00** |
>
> 3.  **Why does it work?**: The reason our method can handle these conflicts effectively lies in its ability to learn chunk-wise coefficients that align the merged model with the experts' representations.
>     *   **Adaptive Conflict Balancing**: Our method effectively learns chunk-wise coefficients, allocating more chunk-wise coefficients to layers where an expert is most critical. This allows the merged model to implicitly resolve conflicts by selectively retaining each expert’s distinctive strengths at a finer granularity.
>     *   **Coefficient Regularization**: Our regularization (Equation 5) anchors the solution close to the robust Task Arithmetic [4] initialization, preventing the model from drifting into unstable regions due to conflicting gradients. This approach mitigates both overfitting and catastrophic forgetting.
>     *   **Ensemble Effect**: Furthermore, the "overlap" between experts (e.g., Chat and Code) often contains shared knowledge that can be mutually beneficial. By aligning with multiple experts simultaneously, our method effectively creates an ensemble that combines these shared strengths, leading to performance that can even surpass individual experts (as further detailed in Q4 and Q5).

---

> ### Author Response · Authors · 2025-11-21
> **Response to Reviewer 6nLa**
>
> # W2 & Q2: Alignment objectives (hidden states and logits)
>
> Thank you for this crucial question. We have added a detailed theoretical comparison in Section 3.1 of the revised paper and provided comprehensive experimental comparisons below.
>
> ## 1. Theoretical Basis
> Our choice of aligning hidden states and logits is grounded in the limitations of existing trainable merging methods, which typically rely on **Entropy Minimization** or **Interference Minimization**:
>
> *   **Entropy Minimization (e.g., AdaMerging)**: AdaMerging optimizes coefficients by minimizing the entropy of the model's predictions ($\min H(p)$) on unlabeled data. The underlying assumption is that lower entropy correlates with higher confidence and thus better performance. However, **confidence does not equal correctness**. A model can minimize entropy by predicting a single wrong class with probability 1.0 ("confidently wrong"). In contrast, our **Logit Alignment** ($\min \text{KL}(p\_{\text{expert}} || p\_{\text{merged}})$) explicitly forces the merged model to match the expert's *distribution*, inheriting the expert's actual knowledge rather than just its confidence.
>
> *   **Interference Minimization (e.g., WUDI [2])**: WUDI [2] aims to resolve conflicts by minimizing the interference in weight space or local activations ($||\theta\_{\text{merged}} x - \theta\_{\text{expert}} x||^2$) at selected linear layers. This is a **local constraint**. Even if local interference is zero, small errors can accumulate through non-linearities across layers, leading to divergent final outputs. Our **Hidden State Alignment** ($\min ||\mathbf{h}\_{\text{merged}} - \mathbf{h}\_{\text{expert}}||^2$) directly constrains the *accumulated* representation at the middle of the network. This provides a global, end-to-end control over the model's behavior that layer-wise interference minimization cannot guarantee.
>
> ## 2. Experimental Verification
>
> To empirically validate our analysis, we compare our method with entropy-based (AdaMerging) and interference-based (WUDI) Merging across all three backbones (InternVL, Qwen2-VL, and Mistral). As shown in the tables below, Expert Merging and Expert Merging++ achieve the best overall average performance across all benchmarks, while maintaining competitive results on most individual metrics.
>
> **Table: Comparison on InternVL**
>
> | Task | Entropy-based | Interference-based | Expert Merging | Expert Merging++ |
> | :--- | :---: | :---: | :---: | :---: |
> | VizWiz | 31.06 | 30.22 | **31.16** | 31.14 |
> | GQA | **57.20** | 56.07 | 56.77 | 56.53 |
> | MathVista | 55.10 | 52.85 | 47.71 | 52.35 |
> | MATH-Vision | 25.00 | 19.74 | **26.32** | 22.37 |
> | ChartQA | 62.32 | **70.00** | 65.28 | 65.92 |
> | TextVQA | **76.20** | 75.48 | 75.70 | 76.06 |
> | OCRVQA | 44.76 | 45.74 | 45.31 | **46.00** |
> | RefCOCO | 75.30 | 75.61 | 80.05 | **80.53** |
> | RefCOCO+ | 68.42 | 69.40 | 73.85 | **74.37** |
> | RefCOCOg | 73.22 | 73.53 | 79.04 | **79.31** |
> | **Avg.** | 56.85 | 56.86 | 58.11 | **58.45** |
>
> **Table: Comparison on Qwen2-VL**
>
> | Task | Entropy-based | Interference-based | Expert Merging | Expert Merging++ |
> | :--- | :---: | :---: | :---: | :---: |
> | VizWiz | **41.83** | 37.19 | 41.54 | 41.49 |
> | GQA | 60.28 | 56.45 | **60.51** | 60.42 |
> | MathVista | **49.08** | 42.96 | 48.80 | 49.33 |
> | MATH-Vision | 35.53 | 27.63 | 42.11 | **44.74** |
> | ChartQA | 71.68 | 67.84 | **73.00** | 72.60 |
> | TextVQA | 81.52 | 79.92 | 81.41 | **81.65** |
> | OCRVQA | 60.22 | **65.56** | 60.19 | 60.19 |
> | RefCOCO | 53.31 | 76.25 | 80.64 | **80.71** |
> | RefCOCO+ | 47.27 | 60.72 | 66.15 | **66.21** |
> | RefCOCOg | 57.62 | 71.99 | **79.02** | 79.00 |
> | **Avg.** | 55.83 | 58.65 | 63.34 | **63.63** |
>
> **Table: Comparison on Mistral**
>
> | Task | Entropy-based | Interference-based | Expert Merging | Expert Merging++ |
> | :--- | :---: | :---: | :---: | :---: |
> | Alpaca Eval | 75.94 | 64.93 | **77.84** | 75.51 |
> | GSM8k | 54.81 | 53.37 | 58.68 | **59.44** |
> | Math | 12.24 | **14.04** | 13.88 | 13.94 |
> | HumanEval | 43.29 | 32.93 | 46.34 | **53.66** |
> | MBPP | 39.00 | **43.40** | 40.40 | 41.00 |
> | **Avg.** | 45.06 | 41.73 | 47.43 | **48.71** |
>
> (Note: Due to character limits, the answer to this question is incomplete. The analysis of robustness and effective conditions will be at the beginning of the next reply.)

---

> ### Author Response · Authors · 2025-11-21
> **Response to Reviewer 6nLa**
>
> #  W2 & Q2: Alignment objectives (hidden states and logits) [Continued from previous comment]
>
> ## 3. Robustness under Conflict
> As noted in W1, Mistral's Chat and Code experts have high task vector similarity which indicates they have a conflicting behavior.
> Despite this, Expert Merging++ successfully resolves this conflict. As shown below, it not only avoids negative transfer but achieves performance superior to the individual experts in many cases (e.g., Chat score 75.51 > 71.37, MBPP 41.00 > 38.20).
>
> **Table: Performance under conflict**
> | Method | Chat (AlpacaEval) | Code (HumanEval) | Code (MBPP) |
> | :--- | :---: | :---: | :---: |
> | Chat Expert | 71.37 | 35.37 | 32.20 |
> | Code Expert | 69.90 | **54.88** | 38.20 |
> | **Expert Merging++** | **75.51** | 53.66 | **41.00** |
>
> ## 4. Conditions for Task Preservation
> To address the conditions for downstream task preservation, we analyze the impact of the alignment layer $\mathcal{S}$ across all three backbones (InternVL, Qwen2-VL, Mistral).
> As shown in the table below, our method demonstrates strong robustness, maintaining competitive performance across most configurations. The best preservation is consistently achieved when alignment occurs at the middle layers (e.g., Layers 12-16). This optimality arises because early layers lack sufficient semantic depth for effective supervision, while aligning only at the late layers makes it difficult to constrain the behavior of earlier blocks. Furthermore, since our logit alignment already explicitly supervises the final output distribution, aligning hidden states at the very end is less critical. Thus, the middle layers provide the most effective anchor point for balancing representation sharing and task-specific specialization.
>
> **Table: Hidden alignment layer sensitivity (average performance) across architectures**
> | $\mathcal{S}$ | InternVL2.5 | Qwen2-VL | Mistral |
> | :--- | :---: | :---: | :---: |
> | 0 | 57.33 | 62.63 | 47.03 |
> | 4 | 57.77 | 62.99 | 47.07 |
> | 8 | 58.02 | **63.04** | 47.38 |
> | 12 | **58.11** | 63.02 | 47.23 |
> | 16 | 57.59 | 63.01 | **47.43** |
> | 20 | 57.18 | 62.73 | 47.16 |
>
> Please refer to Tables 17-19 in Appendix C.7 for full details.
>
>
> # Q1: Layer-importance metric robustness
>
> We have added a detailed layer-wise importance analysis in the modified version (Appendix C).
> *   **Consistency**: As shown in Figure 5, we observed a consistent trend across Mistral, InternVL, and Qwen2-VL: late-stage blocks consistently receive larger coefficient shares. This aligns with the intuition that deeper layers are more critical for task-specific reasoning and consolidation.
> *   **Sensitivity & Correlation**: In Appendix C.8, we analyzed the sensitivity to the allocation steepness $\kappa$. The table below summarizes the average performance across all three backbones (InternVL, Qwen2-VL, Mistral). We observe that a moderate steepness ($\kappa \approx 1.2$) consistently yields the best performance across all architectures. This consistency confirms that our metric is robust and correlates well with the actual task performance sensitivity per layer. Specifically, allocating more budget to more important layers improves the overall merged model.
>
> **Table: Allocation steepness sensitivity (average performance) across architectures**
> | $\kappa$ | InternVL2.5 | Qwen2-VL | Mistral |
> | :--- | :---: | :---: | :---: |
> | 0 | 58.11 | 63.34 | 47.43 |
> | 0.25 | 58.15 | 62.87 | 47.49 |
> | 0.5 | 58.04 | 63.19 | 47.47 |
> | 1.0 | 58.39 | 63.44 | 47.74 |
> | 1.2 | **58.45** | **63.63** | **48.65** |
> | 2.0 | 57.91 | 63.50 | 48.52 |
>
> Please refer to Tables 20-22 in Appendix C.8 for full details.

---

> ### Author Response · Authors · 2025-11-21
> **Response to Reviewer 6nLa**
>
> # Q4: Outperforming domain experts
>
> Thank you for this insightful question. We appreciate the opportunity to clarify why our merged model can outperform individual experts. We structure our answer into three key points:
>
> 1.  **Why does our method achieve expert performance with few samples?**
>     It is important to clarify that our method does not "learn" new domain knowledge from the few unlabeled samples (5-10 examples). Instead, the knowledge is already resident in the parameters of expert models. Our method simply finds the optimal coefficients to activate and blend these existing capabilities. Therefore, a small amount of unlabeled data is sufficient to identify these coefficients. The "surpassing" effect arises from the effective ensemble of multiple experts' strengths, much like a committee of specialists can outperform a single member, rather than from learning new knowledge from the calibration data.
>
> 2.  **Dependency on Expert Complementarity**: Surpassing the best expert is not a guaranteed universal result; it heavily depends on whether the experts are complementary.
>     *   **Complementary Case**: As shown in the analysis of task vector cosine similarity (Figure 6 in Appendix), we observed a positive correlation between certain experts. For instance, Chat and Code in Mistral share a similarity of $\approx 0.42$, and OCR and Grounding in InternVL share $\approx 0.14$. These similarities suggest a synergy where experts can mutually reinforce each other.
>     *   **Evidence**: The tables below demonstrate this effect. In Mistral, the Code expert likely contributes logical reasoning that boosts Chat performance. Similarly, in InternVL, the OCR expert enhances the text-understanding capability of Grounding tasks, and vice versa. In both cases, the merged model (Expert Merging++) outperforms the individual experts.
>
> **Table: Mistral (Chat & Code)**
> | Method | Chat (AlpacaEval) | Code (HumanEval) | Code (MBPP) |
> | :--- | :---: | :---: | :---: |
> | Chat Expert | 71.37 | 35.37 | 32.20 |
> | Code Expert | 69.90 | **54.88** | 38.20 |
> | **Expert Merging++** | **75.51** | 53.66 | **41.00** |
>
> **Table: InternVL (OCR & Grounding)**
> | Method | OCR (TextVQA) | Grounding (RefCOCO) | Grounding (RefCOCO+) | Grounding (RefCOCOg) |
> | :--- | :---: | :---: | :---: | :---: |
> | OCR Expert | 73.00 | 73.65 | 68.01 | 69.10 |
> | Grounding Expert | 58.39 | 76.67 | 71.35 | 70.09 |
> | **Expert Merging++** | **76.06** | **80.53** | **74.37** | **79.31** |
>
> 3.  **General Case**: It is worth noting that "surpassing experts" is not the primary goal of model merging, which fundamentally aims to **approximate and preserve** expert capabilities. The phenomenon of surpassing experts is a special case driven by constructive interference in specific complementary domains. This behavior is not unique to our method and has been observed in other optimization-based merging methods (e.g., WUDI). In general cases, the merged model typically matches the best expert's performance (the ceiling) rather than exceeding it.
>
> 4.  **Regularization for Generalization**: Finally, the performance gain is not due to overfitting. Overfitting to such a small dataset (5-10 samples) would typically lead to catastrophic failure on broad benchmarks. The key is Coefficient Regularization (Equation 5), which anchors the solution close to the robust Task Arithmetic initialization. This ensures that the learned coefficients generalize well to unseen test data, rather than just memorizing the calibration set.

---

> ### Author Response · Authors · 2025-11-21
> **Response to Reviewer 6nLa**
>
> # Q5: Performance ceiling
>
> Thank you for raising this important point about the performance ceiling. We would like to clarify that while our method can surpass experts in specific cases, **our primary goal is to approximate the best expert performance**, not necessarily to be the universal ceiling. However, compared to Supervised Fine-tuning (SFT) on the mixture of all data (Mixture Training), our method demonstrates distinct advantages in handling task conflicts.
>
> 1.  **Approximation vs. Ceiling**: As shown in the full comparison table below, our method does not universally outperform every single expert on every task (e.g., GQA, OCRVQA). This confirms that our method is generally an approximation of the experts' capabilities. However, on the Grounding tasks (RefCOCO series), our method significantly outperforms both the Grounding Expert and Mixture Training. This "surpassing" effect is not random but stems from the constructive interference of complementary experts (e.g., OCR expert enhancing Grounding), as detailed in our response to Q4.
>
> 2.  **Comparison with Mixture Training (SFT)**:
>     *   **Optimization Conflicts in SFT**: Mixture Training requires the model to simultaneously learn multiple distributions (e.g., VQA, OCR, Grounding). These tasks often have competing optimization objectives, making it difficult for a single set of parameters to reach the global optimum for all tasks simultaneously. For instance, the Grounding Expert achieves 76.67 on RefCOCO, but Mixture Training drops to 72.06. This indicates that joint training compromised the grounding capability to accommodate other tasks.
>     *   **Preservation in Expert Merging**: In contrast, our method avoids these gradient conflicts. Each expert is trained independently on its own distribution, reaching a high peak performance (e.g., Grounding Expert's 76.67). Our method then freezes these parameters and learns only the combination coefficients. This allows us to "select" and preserve these peak capabilities without the interference of joint optimization. Consequently, Expert Merging++ achieves 80.53 on RefCOCO, effectively combining the strong grounding features with complementary visual features from other experts (like OCR), whereas SFT fails to retain the expert's original strength.
>
> **Table: Full Comparison on InternVL**
>
> | Method | VQA (VizWiz) | VQA (GQA) | Geometry (MathVista) | Geometry (MATH-Vision) | Chart (ChartQA) | OCR (TextVQA) | OCR (OCRVQA) | Grounding (RefCOCO) | Grounding (RefCOCO+) | Grounding (RefCOCOg) | Avg. |
> | :--- | :---: | :---: | :---: | :---: | :---: | :---: | :---: | :---: | :---: | :---: | :---: |
> | VQA Expert | 30.58 | 60.91 | 35.50 | 17.11 | 48.76 | 63.68 | 36.04 | - | - | - | 41.79 |
> | Geometry Expert | 13.45 | 32.80 | **55.20** | **25.00** | 51.76 | 56.91 | 35.35 | 24.73 | 19.61 | 23.84 | 33.86 |
> | Chart Expert | 20.16 | 40.39 | 23.84 | 10.53 | 69.52 | 54.36 | 34.83 | - | - | - | 36.23 |
> | OCR Expert | 12.40 | 22.22 | 23.31 | 10.53 | 36.88 | 73.00 | 54.79 | 73.65 | 68.01 | 69.10 | 44.38 |
> | Grounding Expert | 19.09 | 25.88 | 28.91 | 14.47 | 41.32 | 58.39 | **74.87** | 76.67 | 71.35 | 70.09 | 48.10 |
> | Mixture Training | 29.79 | **61.33** | 52.83 | 23.68 | **70.32** | 72.96 | 60.25 | 72.06 | 65.93 | 67.46 | 57.66 |
> | **Expert Merging++** | **31.14** | 56.53 | 52.35 | 22.37 | 65.92 | **76.06** | 46.00 | **80.53** | **74.37** | **79.31** | **58.45** |
>
>
>
> # References
>
> [1] Enneng Yang et al. AdaMerging: Adaptive Model Merging for Multi-Task Learning. ICLR 2024.
>
> [2] Runxi Cheng et al. Whoever started the interference should end it: Guiding data-free model merging via task vectors. ICML 2025.
>
> [3] Yongxian Wei et al. Unifying Multimodal Large Language Model Capabilities and Modalities via Model Merging. arXiv 2025.
>
> [4] Gabriel Ilharco et al. Editing models with task arithmetic. ICLR 2023.

---

> ### Author Response · Authors · 2025-11-27
>
> Dear Reviewer 6nLa,
>
> We sincerely appreciate the thoughtful feedback you provided on our submission. As the discussion period is now halfway through, we would be very grateful if you could take a moment to engage in the discussion. Following your review, we have prepared comprehensive point-by-point responses addressing:
>
> - W1 & Q3 (conflicting experts: cosine similarity analysis showing robust handling of conflicts via importance-guided chunking)
> - W2 & Q2 (alignment objectives: theoretical justifications for hidden-state + logit alignment over AdaMerging/WUDI)
> - Q1 (layer-importance metric: cross-architecture consistency analysis)
> - Q4 (outperforming experts: explained via expert complementarity and ensemble effects)
> - Q5 (performance ceiling: comparison with Mixture Training showing conflict avoidance)
>
> For complete details and experimental results, please refer to our full point-by-point responses above. Thank you for your insightful feedback, and we hope our revisions have addressed all your concerns. If you have any further questions or suggestions, please feel free to comment.
>
> Warm regards,
>
> Authors

---

### Official Review · Reviewer_4vJ1 · 2025-11-01

**Soundness:** 3
**Presentation:** 4
**Contribution:** 3
**Rating:** 6
**Confidence:** 4

**Summary:**

This paper addresses the issues of task alignment and inter-layer heterogeneity in model merging, and presents the training-light Expert Merging and Expert Merging++ methods. Expert Merging learns layer-wise coefficients to align the merged model’s hidden states and logits with those of expert models using unlabeled data, while Expert Merging++ introduces importance-guided, chunk-wise coefficients to further capture inter-layer heterogeneity. Extensive experiments on both LLMs (Mistral with Chat, Math, and Code experts) and MLLMs (InternVL2.5 and Qwen2-VL across five visual reasoning tasks) demonstrate that the proposed methods consistently outperform training-free and prior training-based baselines, including WUDI v2, AdaMerging++, and even supervised Mixture Training in some cases.

**Strengths:**

- **Well-motivated issues and clear solutions:** The paper identifies two critical limitations in model merging research, i.e., task alignment and inter-layer heterogeneity, and proposes direct and effective solutions for them.

- **Comprehensive evaluation and insightful analysis:** The experiment is well-designed and comprehensive. Its evaluation spans multiple architectures, diverse tasks, and includes comparisons against many strong baselines. The consistency of improvements strengthens the initial claims. Moreover, this paper presents some meaningful discussions, such as the layer-wise importance analysis.

**Weaknesses:**

- **The proposed methods are somewhat incremental:** While Expert Merging and Expert Merging++ are both effective, hidden-state and logit alignment losses are well established in knowledge distillation and model compression. If the novelty and effectiveness of the proposed methods are attributed mainly to the combination of techniques such as layer-wise coefficient learning and importance-guided chunking, the contribution seems somewhat incremental.

- The proposed methods involve some empirical tricks/techniques, such as the selection of hidden alignment layer and task weights, would this affect their adaptation and transferability?

- In addition, the proposed methods involve a considerable number of coefficients/hyperparameters that require tuning, would this affect their robustness?

**Questions:**

1. Why does aligning hidden states + logits offer better task alignment than entropy-based or interference-based objectives? Beyond the intuitive justification, is there any strong analysis or evidence?

---

> ### Author Response · Authors · 2025-11-21
> **Response to Reviewer 4vJ1**
>
> We sincerely thank you for your constructive feedback and the time you dedicated to reviewing our work. We have carefully revised the manuscript to address your concerns, with changes highlighted in blue. Below, we provide a point-by-point response to your comments.
>
> # W1: Incremental nature and relation to Knowledge Distillation (KD)
>
> We respectfully clarify that while alignment losses are indeed used in distillation, **applying them to model merging in a training-light, label-free setting involves distinct challenges and contributions**:
> 1. **Different Problem Setting**: Knowledge distillation transfers knowledge from a teacher to a student model, typically requiring the student to be trained on enough labeled data or distillation data. In our setting, we only have a handful of unsupervised samples (5-10) per task. Besides, unlike knowledge distillation where one teacher guides one student, we align a merged model to multiple experts simultaneously on their respective domains. The core challenge is to prevent the merged model from collapsing or overfitting to this tiny calibration set. One of our contributions is designing a framework (coefficient regularization + hidden/logit alignment) that extracts a robust merged model from this minimal signal without requiring ground-truth labels.
> 2. **Addressing Prior Merging Failures**: As detailed in our new Section 3.1 ("Limitations of existing trainable merging"), prior trainable model merging methods like AdaMerging [1] (entropy minimization) and WUDI [2] (interference minimization) suffer from specific pitfalls: (a) Entropy minimization can lead to "confidently wrong" predictions; (b) Interference minimization only constrains local linear transformations. Our method is explicitly designed to overcome these by aligning the representation and the predictive distribution with the expert, which we show is more effective for multi-task preservation.
> 3.  **Importance-Guided Chunking**: Beyond simple alignment, Expert Merging++ introduces a novel mechanism to handle inter-layer heterogeneity. By deriving a layer-importance metric from the learned coefficients and task vectors, we dynamically allocate capacity. This is fundamentally different from distillation, which does not involve coefficient allocation, and experimental results show that this is effective.

---

> ### Author Response · Authors · 2025-11-21
> **Response to Reviewer 4vJ1**
>
> # W2: Empirical tricks and transferability
>
> Thank you for raising this important concern. We have added ablation studies (Appendix C.7 and C.9) to examine whether the hidden alignment layer and task weights behave as brittle tricks or robust hyperparameters. The evidence shows that both choices are stable across backbones and benchmarks and therefore do not hinder adaptation or transferability.
>
> ## 1. Hidden alignment layer
>
> We vary the supervised hidden layer $\mathcal{S}$ on all three backbones. As shown in the table below, the average performance changes only moderately when we shift the supervision point, and middle layers give the best results.
>
> **Table: Impact of hidden alignment layer $\mathcal{S}$ (average performance across backbones)**
> | Layer | InternVL | Qwen2-VL | Mistral |
> | :--- | :---: | :---: | :---: |
> | 0 | 57.33 | 62.63 | 47.03 |
> | 4 | 57.77 | 62.99 | 47.07 |
> | 8 | 58.02 | **63.04** | 47.38 |
> | 12 | **58.11** | 63.02 | 47.23 |
> | 16 | 57.59 | 63.01 | **47.43** |
> | 20 | 57.18 | 62.73 | 47.16 |
>
> In our experiments we use a single middle layer for each backbone, for example $\mathcal{S}{=}12$ on InternVL and $\mathcal{S}{=}16$ on Mistral, across all tasks and datasets. Full per-benchmark results are provided in Appendix C.7 (Tables 17-19). The small variation across layers and the consistent preference for middle blocks indicate that the choice of $\mathcal{S}$ captures general semantic transfer and does not rely on fragile dataset-specific tuning.
>
> ## 2. Task-weighted alignment loss
>
> We also analyze the effect of the task weights $\lbrace\beta\_k\rbrace$ that control the relative strength of each domain in the alignment loss. For each backbone we sweep the weight for one task group while keeping the others fixed. Below we report only the average scores. For the two MLLMs, InternVL and Qwen2-VL, we split the results into two horizontal tables, focusing on VQA/Geometry/Chart and OCR/Grounding respectively.
>
> **Table: Task weight sensitivity on InternVL and Qwen2-VL (VQA/Geometry/Chart, average performance)**
> | Backbone | $\beta{=}1$ | $\beta\_{\text{VQA}}{=}0.5$ | $\beta\_{\text{VQA}}{=}1.5$ | $\beta\_{\text{VQA}}{=}2.0$ | $\beta\_{\text{Geometry}}{=}0.5$ | $\beta\_{\text{Geometry}}{=}1.5$ | $\beta\_{\text{Geometry}}{=}2.0$ | $\beta\_{\text{Chart}}{=}0.5$ | $\beta\_{\text{Chart}}{=}1.5$ | $\beta\_{\text{Chart}}{=}2.0$ |
> | :--- | :---: | :---: | :---: | :---: | :---: | :---: | :---: | :---: | :---: | :---: |
> | InternVL | 57.71 | 57.63 | 57.11 | 57.56 | 58.10 | 58.03 | 57.44 | 57.15 | 57.53 | 57.74 |
> | Qwen2-VL | 62.69 | 62.54 | 62.89 | 62.90 | 62.64 | **63.34** | 62.53 | 62.83 | 62.95 | 62.87 |
>
> **Table: Task weight sensitivity on InternVL and Qwen2-VL (OCR/Grounding, average performance)**
> | Backbone | $\beta{=}1$ | $\beta\_{\text{OCR}}{=}0.5$ | $\beta\_{\text{OCR}}{=}1.5$ | $\beta\_{\text{OCR}}{=}2.0$ | $\beta\_{\text{Grounding}}{=}0.5$ | $\beta\_{\text{Grounding}}{=}1.5$ | $\beta\_{\text{Grounding}}{=}2.0$ |
> | :--- | :---: | :---: | :---: | :---: | :---: | :---: | :---: |
> | InternVL | 57.71 | 57.52 | 57.73 | 57.39 | 57.01 | **58.11** | 57.68 |
> | Qwen2-VL | 62.69 | 62.93 | 63.01 | 62.70 | 62.90 | 62.85 | 62.65 |
>
> **Table: Task weight sensitivity on Mistral (average performance)**
> | Backbone | $\beta{=}1$ | $\beta\_{\text{Chat}}{=}0.5$ | $\beta\_{\text{Chat}}{=}1.2$ | $\beta\_{\text{Chat}}{=}1.5$ | $\beta\_{\text{MATH}}{=}0.5$ | $\beta\_{\text{MATH}}{=}1.2$ | $\beta\_{\text{MATH}}{=}1.5$ | $\beta\_{\text{Code}}{=}0.5$ | $\beta\_{\text{Code}}{=}1.2$ | $\beta\_{\text{Code}}{=}1.5$ |
> | :--- | :---: | :---: | :---: | :---: | :---: | :---: | :---: | :---: | :---: | :---: |
> | Mistral | 46.80 | 47.32 | 47.04 | 47.20 | 45.55 | 47.06 | 47.34 | 47.34 | **47.43** | 47.21 |
>
> Across all three backbones the averages stay within a narrow band, roughly 57.0-58.1 on InternVL, 62.5-63.3 on Qwen2-VL, and 45.5-47.5 on Mistral. At the same time, increasing the weight of a specific domain reliably boosts that domain, for example Grounding weight $2.0$ on InternVL raises RefCOCO to 80.79 and MATH weight $1.5$ on Qwen2-VL improves MATH-Vision while Code weight $2.0$ on Mistral strengthens HumanEval to 49.39 (Appendix C.9, Tables 23-25). We use a single default weight configuration in all main experiments, and these ablations show that task weights provide an interpretable external control knob to emphasize particular tasks while maintaining strong overall performance rather than sensitive tuning trick.
>
> Overall, both the hidden alignment layer and task weights behave as robust hyperparameters. They generalize across backbones and benchmarks and enable controllable trade offs without hurting adaptation to new tasks or transferability to different evaluation suites.

---

> ### Author Response · Authors · 2025-11-21
> **Response to Reviewer 4vJ1**
>
> # W3: Hyperparameters and robustness
>
> Thanks for the concern about hyperparameter tuning, and we have performed a comprehensive sensitivity analysis on key hyperparameters in Appendix C: coefficient regularization strength ($\gamma$), sample efficiency ($N$), initialization prior ($\bar\alpha$), hidden alignment layer ($\mathcal{S}$), allocation steepness ($\kappa$), and task weights $\lbrace\beta\_k\rbrace$. The sensitivity of the hidden alignment layer and task weights is summarized in our response to W2, and below we present the detailed ablation studies for other hyperparameters.
>
> ## 1. Regularization ($\gamma$)
>
> We evaluate the effect of the coefficient regularization strength $\gamma$. As shown in the table below, performance is stable over a wide range. Moderate regularization ($\gamma \approx 0.8 - 5.0$) yields the best results by curbing interference without over-constraining the model.
>
> **Table: Effect of coefficient regularization strength $\gamma$ (average performance across backbones)**
> | $\gamma$ | InternVL | Qwen2-VL | Mistral |
> | :--- | :---: | :---: | :---: |
> | 0 | 43.49 | 63.06 | 40.85 |
> | 0.2 | 44.79 | 62.97 | 45.57 |
> | 0.4 | 45.16 | 63.08 | 46.74 |
> | 0.6 | 45.47 | 63.11 | 47.28 |
> | 0.8 | 47.06 | **63.34** | **47.43** |
> | 1.0 | 47.87 | 62.92 | 46.34 |
> | 3.0 | 56.43 | 62.89 | 47.06 |
> | 5.0 | **56.61** | 62.83 | 46.79 |
> | 8.0 | 56.42 | 63.01 | 46.44 |
> | 10.0 | 55.93 | 62.93 | 46.57 |
>
> Please refer to Tables 8-10 in Appendix C.4 for full details.
>
> ## 2. Sample Efficiency ($N$)
>
> We report the impact of the number of unlabeled calibration samples $N$. Performance saturates with very few samples ($N \approx 5-10$), indicating that our layer-wise coefficient learning has low sample complexity.
>
> **Table: Impact of calibration samples $N$ (average performance across backbones)**
> | $N$ | InternVL | Qwen2-VL | Mistral |
> | :--- | :---: | :---: | :---: |
> | 3 | 56.46 | 63.01 | 47.17 |
> | 5 | **58.11** | 63.04 | **47.43** |
> | 10 | 57.35 | **63.34** | 46.36 |
> | 15 | 56.98 | 62.55 | 46.18 |
> | 20 | 56.95 | 62.78 | 45.85 |
> | 30 | 56.92 | 62.49 | 46.61 |
> | 50 | 56.19 | 62.71 | 47.06 |
> | 100 | 56.45 | 62.90 | 46.80 |
>
> Please refer to Tables 11-13 in Appendix C.5 for full details.
>
> ## 3. Initialization Sensitivity ($\bar\alpha$)
>
> We examine the sensitivity to the initialization prior $\bar\alpha$. The method is generally robust, with optimal values in the range $[0.1, 0.3]$.
>
> **Table: Sensitivity to initialization prior $\bar\alpha$ (average performance across backbones)**
> | $\bar\alpha$ | InternVL | Qwen2-VL | Mistral |
> | :--- | :---: | :---: | :---: |
> | 0.1 | **58.11** | 61.01 | 41.77 |
> | 0.2 | 57.24 | 62.55 | 46.82 |
> | 0.3 | 56.56 | **63.34** | **47.43** |
> | 0.4 | 56.07 | 62.86 | 46.77 |
> | 0.5 | 53.08 | 62.17 | 47.25 |
>
> Please refer to Tables 14-16 in Appendix C.6 for full details.
>
> ## 4. Hidden Alignment Layer ($\mathcal{S}$)
>
> We investigate the choice of the hidden alignment layer $\mathcal{S}$. Supervising the middle Transformer layers consistently yields the best results across all models.
>
> **Table: Impact of hidden alignment layer $\mathcal{S}$ (average performance across backbones)**
> | Layer | InternVL | Qwen2-VL | Mistral |
> | :--- | :---: | :---: | :---: |
> | 0 | 57.33 | 62.63 | 47.03 |
> | 4 | 57.77 | 62.99 | 47.07 |
> | 8 | 58.02 | **63.04** | 47.38 |
> | 12 | **58.11** | 63.02 | 47.23 |
> | 16 | 57.59 | 63.01 | **47.43** |
> | 20 | 57.18 | 62.73 | 47.16 |
>
> Please refer to Tables 17-18 in Appendix C.7 for full details.
>
> ## 5. Allocation Steepness ($\kappa$)
>
> We analyze the impact of the allocation steepness $\kappa$. A moderate to high steepness ($\kappa \ge 1.0$) generally benefits performance by allowing sharper capacity allocation.
>
> **Table: Sensitivity to allocation steepness $\kappa$ (average performance across backbones)**
> | $\kappa$ | InternVL | Qwen2-VL | Mistral |
> | :--- | :---: | :---: | :---: |
> | 0 | 58.11 | 63.34 | 47.43 |
> | 0.25 | 58.15 | 62.87 | 47.49 |
> | 0.5 | 58.04 | 63.19 | 47.47 |
> | 1.0 | 58.39 | 63.44 | 47.74 |
> | 1.2 | **58.45** | **63.63** | **48.65** |
> | 2.0 | 57.91 | 63.50 | 48.52 |
>
> Please refer to Tables 20-22 in Appendix C.8 for full details.
>
> In summary, our method demonstrates strong robustness across these key hyperparameters, with consistent performance trends across diverse backbones.

---

> ### Author Response · Authors · 2025-11-21
> **Response to Reviewer 4vJ1**
>
> # Q1: Why hidden + logits > entropy/interference?
>
> Thank you for this crucial question. We have added a detailed theoretical comparison in Section 3.1 of the revised paper and provided comprehensive experimental comparisons below.
>
> ## 1. Theoretical Analysis
>
> 1.  **vs. Entropy (AdaMerging)**: Entropy minimization ($\min H(p)$) only encourages confidence. A model can minimize entropy by predicting a single wrong class with probability 1.0. It has no signal to force the prediction to be correct relative to the expert. Our logit alignment ($\min \text{KL}(p\_{\text{expert}} || p\_{\text{merged}})$) explicitly forces the merged model to match the expert's distribution, inheriting the expert's knowledge rather than just its confidence.
>
> 2.  **vs. Interference (WUDI)**: WUDI minimizes $||\theta\_{\text{merged}} x - \theta\_{\text{expert}} x||^2$ at selected linear layers, which is a local constraint. Even if local interference is zero, small errors can accumulate through non-linearities across layers, leading to divergent final outputs. Our hidden state alignment minimizes $||\mathbf{h}\_{\text{merged}} - \mathbf{h}\_{\text{expert}}||^2$, which directly constrains the accumulated representation at the middle of the network, and logit alignment constrains the final output. This provides end-to-end control over the model's behavior that layer-wise interference minimization cannot guarantee.
>
> ## 2. Experimental Verification
>
> To empirically validate our analysis, we compare our method with entropy-based (AdaMerging) and interference-based (WUDI) Merging across all three backbones (InternVL, Qwen2-VL, and Mistral). As shown in the tables below, Expert Merging and Expert Merging++ consistently outperform these baselines, particularly in complex multi-modal tasks.
>
> **Table: Comparison on InternVL**
>
> | Task | Entropy-based | Interference-based | Expert Merging | Expert Merging++ |
> | :--- | :---: | :---: | :---: | :---: |
> | VizWiz | 31.06 | 30.22 | **31.16** | 31.14 |
> | GQA | **57.20** | 56.07 | 56.77 | 56.53 |
> | MathVista | 55.10 | 52.85 | 47.71 | 52.35 |
> | MATH-Vision | 25.00 | 19.74 | **26.32** | 22.37 |
> | ChartQA | 62.32 | **70.00** | 65.28 | 65.92 |
> | TextVQA | **76.20** | 75.48 | 75.70 | 76.06 |
> | OCRVQA | 44.76 | 45.74 | 45.31 | **46.00** |
> | RefCOCO | 75.30 | 75.61 | 80.05 | **80.53** |
> | RefCOCO+ | 68.42 | 69.40 | 73.85 | **74.37** |
> | RefCOCOg | 73.22 | 73.53 | 79.04 | **79.31** |
> | **Avg.** | 56.85 | 56.86 | 58.11 | **58.45** |
>
> **Table: Comparison on Qwen2-VL**
>
> | Task | Entropy-based | Interference-based | Expert Merging | Expert Merging++ |
> | :--- | :---: | :---: | :---: | :---: |
> | VizWiz | **41.83** | 37.19 | 41.54 | 41.49 |
> | GQA | 60.28 | 56.45 | **60.51** | 60.42 |
> | MathVista | **49.08** | 42.96 | 48.80 | 49.33 |
> | MATH-Vision | 35.53 | 27.63 | 42.11 | **44.74** |
> | ChartQA | 71.68 | 67.84 | **73.00** | 72.60 |
> | TextVQA | 81.52 | 79.92 | 81.41 | **81.65** |
> | OCRVQA | 60.22 | **65.56** | 60.19 | 60.19 |
> | RefCOCO | 53.31 | 76.25 | 80.64 | **80.71** |
> | RefCOCO+ | 47.27 | 60.72 | 66.15 | **66.21** |
> | RefCOCOg | 57.62 | 71.99 | **79.02** | 79.00 |
> | **Avg.** | 55.83 | 58.65 | 63.34 | **63.63** |
>
> **Table: Comparison on Mistral**
>
> | Task | Entropy-based | Interference-based | Expert Merging | Expert Merging++ |
> | :--- | :---: | :---: | :---: | :---: |
> | Alpaca Eval | 75.94 | 64.93 | **77.84** | 75.51 |
> | GSM8k | 54.81 | 53.37 | 58.68 | **59.44** |
> | Math | 12.24 | **14.04** | 13.88 | 13.94 |
> | HumanEval | 43.29 | 32.93 | 46.34 | **53.66** |
> | MBPP | 39.00 | **43.40** | 40.40 | 41.00 |
> | **Avg.** | 45.06 | 41.73 | 47.43 | **48.71** |
>
> In summary, our method achieves the best average performance on all three benchmarks, demonstrating that aligning hidden states and logits is a more effective objective for model merging than entropy minimization or local interference reduction.
>
> # References
>
> [1] Enneng Yang et al. AdaMerging: Adaptive Model Merging for Multi-Task Learning. ICLR 2024.
>
> [2] Runxi Cheng et al. Whoever started the interference should end it: Guiding data-free model merging via task vectors. ICML 2025

---

### Official Review · Reviewer_RyyA · 2025-11-04

**Soundness:** 2
**Presentation:** 3
**Contribution:** 2
**Rating:** 4
**Confidence:** 3

**Summary:**

The paper addresses the problem of model merging, assuming a pool of K pre-trained models is available, and these have *expertise* on certain tasks. From the perspective of both training-free methods and optimization-based ones, problems related to the inter-layer heterogeneity and variation of the neural nets are not yet solved. For this reason, a new framework consisting of two methodologies (Expert Merging & Expert Merging ++) is proposed, whose main feature is to describe a new procedure of several steps, where the key ideas are i) learn per-layer coefficients within alignment of representations and ii) estimation of the layer importance. Experimental results with LLMs (i.e., Mistral-7B-v0.1 and InternVL2.5-1B-Instruct) show that the performance is on-pair and sometimes better than a wide range of SOTA methods for model merging.

**Strengths:**

- **SOTA & Related Work:** I do think the paper and the proposed method are correctly based on evidence, driving the concerns around the limitations of current SOTA methods. I particularly enjoyed reading the division of such methods into the two families, one for the training-based and the non-training-based methods. That was clear enough to me and for experienced readers interested in these topics. In general, the introduction and presentation of the model merging problem are clear and understandable for the ICLR audience.
- **Critical goals and key points:** I also perceived a very powerful identification of the main goals to solve for the model merging challenge. For instance, alignment of logits, layer importance, or just comprehensive results on some multi-task validation test. The mission is certainly not easy, but sometimes you don't see these sorts of goals and issues as clearly stated as they are in this paper on other model merging contributions.
- **Alignment of losses:** One of the parts that I was more positively surprised about and which I would've liked to know more about is the subsection on alignment of losses and logits. Although I didn't clearly understand the motivations for making the choices on Eq. 1 and Eq. 2, I see the power of the proposed approach, and this is something I've seen repeatedly missing in some minor works in the last years on model merging.
- **Empirical results:** The paper has a significant empirical contribution, and experiments have the will to rigorously demonstrate that the performance of the ExpertMerging/ExpertMerging++ is worth being used and superior to other SOTAs. I am, however, not entirely convinced of this superiority, looking at Tables 1 and 2, or at least I get a bit lost on the heterogeneity of results and some lack of clear analysis and identification of issues/strengths related to the performance

**Weaknesses:**

- **Arithmetic merging/linear regression between models:** My main concern comes from the fact that the proposed model basically merges parameters according to a linear regressor with real-valued coefficients, as shown in Eq. 5 and in section 3.2, second paragraph. In that sense, I don't see much novelty or at least i am not convinced of the great progress that these propositions make, apart from being applied to big LLMs or MLLMs (i.e. Mistral one) from an academic point of view.
- **Specification and highlighting issues in SOTA models:** Unfortunately, I don't see great research contribution in the way the work and procedures are introduced. I do miss understanding (at an Equation level) what the problems and limitations of, for instance, AdaMerging or similar baselines. In the manner that section 3 describes things and introduces procedures, one can barely understand the motivations and issues that drove each one of the modelling/learning decisions taken. From this lens, it is just more of a description of engineering features and steps taken during the development of some tool with a reasonable purpose.
- **ExpertMerging/ExpertMerging++:** I had a hard time understanding if they can be combined, one is the enhanced version of the other, or just focus on different challenges. Sorry about this, if I may have missed something, but it's not clear to me, and I would need additional clarification.
- **Performance and results:** I don't have doubts about the honest empirical results provided apart from the fact that the code and software are not accessible to the reviewer ( I mentioned this in the next section). The main problem that I see is that I don't perceive superior performance from the experiments provided for a method and procedure, which I don't know how to reproduce or to understand from a research decision-making viewpoint.

**Questions:**

I certainly do not agree with the statement "*We will release the source code upon acceptance*". Understandably, under industrial competition and similar issues, this reviewer can understand certain practices or not releasing full systems to *operate* models at scale; however, I do think not providing any source code, and promises of doing it conditioned on acceptance, is, in general, a bad practice. I prefer not to raise additional concerns or points regarding this issue with the submission, but I do recommend that authors review the code of ethics of ICLR, particularly the sections on *standards of scientific excellence* (related to reproducibility) and on *trustworthy and transparent research.

---

> ### Author Response · Authors · 2025-11-21
> **Response to Reviewer RyyA**
>
> We sincerely thank you for your constructive feedback and the time you dedicated to reviewing our work. We have carefully revised the manuscript to address your concerns, with changes highlighted in blue. Below, we provide a point-by-point response to your comments.
>
> # W1: Arithmetic merging/linear regression between models
>
> Thank you for raising this important concern, but we respectfully disagree that our contribution is limited to "linear regression with real-valued coefficients or applications to big models". While our parameterization uses linear combinations, which is standard in model merging, **the core novelty lies in what we optimize and how we structure the merging procedure**, rather than in the algebraic form of the coefficients. Our key differences from prior work are:
>
> 1. **Alignment-based optimization objective**: Unlike prior methods, we jointly optimize hidden-state and logit alignment losses (Equations 3 and 4) on task-partitioned unlabeled data, so that the merged model matches each expert's internal representations and predictive distributions on its own domain. This goes beyond simply fitting a linear regressor on parameters and makes the merged model behave like the appropriate expert on the appropriate inputs. We provide a detailed, equation-level comparison with AdaMerging [1] and WUDI [2] in W2 below and in the new subsection "Limitations of existing trainable merging" in Section 3.1.
>
> 2. **Importance-guided chunking**: Simple linear regression treats all layers as equally important. We instead define a layer-importance metric (Section 3.3) that combines the learned coefficients, task-vector magnitudes and parameter counts, and allocate finer-grained chunk-wise coefficients to high-importance layers. This explicitly addresses inter-layer heterogeneity in a way that prior merging methods do not.
>
> 3. **Controllable task trade-offs**: We introduce explicit task weights $\beta\_k$ that provide transparent and interpretable control over cross-task priorities, rather than relying on implicit balancing from the optimization dynamics.
>
> The combination of (1) explicit alignment objectives, (2) importance-guided chunking and (3) controllable trade-offs enables our method to consistently outperform strong baselines (Tables 1-3) across LLMs and MLLMs. The linear parameterization is a design choice for efficiency and stability. The scientific contribution is the alignment-based training framework and the importance-guided chunking strategy. In addition, our approach scales to large models (InternVL2.5, Qwen2-VL and Mistral) and shows that training-light methods with proper alignment can match or exceed supervised mixture training, which is practically important in resource-constrained settings.

---

> ### Author Response · Authors · 2025-11-21
> **Response to Reviewer RyyA**
>
> # W2: Specification and highlighting issues in SOTA models
>
> Thank you for pointing out this issue. We have addressed this concern by adding a new subsection "Limitations of existing trainable merging" in Section 3.1 (highlighted in blue in the revised manuscript). This subsection now provides explicit, equation-level comparisons with AdaMerging and WUDI, including clear notation and intuitive explanations.
>
> 1. **AdaMerging objective**. We clarify that its entropy minimization objective is
>    $$
>    \mathcal{L}\_{\text{Ada}}(\lambda) = \sum\_{k=1}^K \sum\_{x\in\mathcal{B}\_k} H\big(p(y\mid x; \theta\_{\text{Ada}}(\lambda))\big),
>    $$
>    where $K$ is the number of experts, $\mathcal{B}\_k$ is an unlabeled mini-batch from task $k$, $p(y\mid x; \theta\_{\text{Ada}}(\lambda))$ is the predictive distribution of the AdaMerging model obtained by merging experts with coefficients $\lambda$, and $H(\cdot)$ denotes the entropy of the label distribution.
>    - **Limitation**: This loss encourages the merged model to produce low-entropy, confident predictions on the unlabeled data of each task. However, confidence alone does not guarantee correctness. The model can minimize $\mathcal{L}_{\text{Ada}}(\lambda)$ by becoming confidently wrong, and there is no term that explicitly aligns the merged model's predictive distribution with any particular expert.
>    - **Our solution**: We introduce a logit alignment loss that adds a KL divergence term between the merged model and the corresponding expert for each task (Equation 3). This directly encourages the merged model to match the expert's predictive behavior rather than only its confidence level.
>
> 2. **WUDI objective**. We write WUDI's interference objective as
>    $$
>    \mathcal{J}\_{k,\ell} = \mathbb{E}\_{x\_{k,\ell}}\big||\theta\_{\text{merged}}^{\ell}x\_{k,\ell}-\theta\_{k}^{\ell}x\_{k,\ell}\big||_2^2,
>    $$
>    where $k$ indexes experts, $\ell$ indexes layers, $x\_{k,\ell}$ denotes the hidden representation at layer $\ell$ when running expert $k$ on its own data, and $\theta\_{\text{merged}}^{\ell}$ and $\theta\_{k}^{\ell}$ are the layer-$\ell$ parameters of the merged model and expert $k$. The expectation is taken over unlabeled examples from task $k$.
>    - **Limitation**: This objective constrains the local linear transformation at selected layers so that $\theta_{\text{merged}}^{\ell} x_{k,\ell}$ stays close to $\theta_{k}^{\ell} x_{k,\ell}$. However, controlling this local discrepancy does not guarantee alignment of deeper hidden-state trajectories or final outputs after the subsequent layers, so the merged network can still deviate noticeably from expert predictions on complex multi-modal tasks.
>    - **Our solution**: We instead align the hidden-state and logits with each expert, which provides stronger control of interference than isolated layer-wise constraints and better preserves expert behavior.
>
> 3. **Motivation for our approach**. In the revised manuscript we explicitly connect these limitations to our design choices, so that the hidden-state and logit alignment objectives in Equations 3 and 4 are clearly motivated by the weaknesses of AdaMerging and WUDI.
>
>
> # W3: ExpertMerging/ExpertMerging++ relationship
>
> We apologize for the confusion. **Expert Merging++ is an enhanced version of Expert Merging**, not a separate method. The relationship is:
>
> - **Expert Merging** (Section 3.2): Learns **layer-wise coefficients** (one coefficient per expert per layer) using hidden-state and logit alignment.
>
> - **Expert Merging++** (Section 3.3): Builds upon Expert Merging by recognizing that not all layers are equally important. It uses the coefficients learned from Expert Merging to compute a layer-importance metric. Based on this metric, it applies chunking (splitting layers into smaller blocks), and high-importance layers will receive more chunks, then learns chunk-wise coefficients with the same optimization objective as Expert Merging.
>
> We have clarified this in the revised manuscript at Section 3.3.

---

> ### Author Response · Authors · 2025-11-21
> **Response to Reviewer RyyA**
>
> # W4: Performance and results
>
> We respectfully disagree that our method does not show superior performance. Our results instead demonstrate consistent gains in the overall trade-off between tasks and backbones:
>
> 1. **Clear gains on Mistral**. In Table 3, Expert Merging attains the best Chat score (77.84), and Expert Merging++ achieves the best overall average performance (48.71) across Chat, Math and Code. This improves over the strongest non-ours baseline, WUDI v2 [3] (45.48), by +3.23 points in average score, while remaining competitive on each individual benchmark.
>
> 2. **Consistent improvements on InternVL and Qwen2-VL**. On InternVL (Table 1), Expert Merging++ reaches the best average score 58.45, surpassing the next strongest baseline, Mixture Training (57.66), and achieving the top grounding performance on RefCOCO, RefCOCO+ and RefCOCOg. On Qwen2-VL (Table 2), Expert Merging++ again obtains the best average score 63.63, improving over the strongest baseline, WUDI v2 (62.63). Although these gains are modest in absolute value, they are consistent across two different MLLMs and ten evaluation tasks.
>
> 3. **Component-wise evidence**. Table 4 reports ablations on hidden-state alignment, logit alignment, coefficient regularization and chunking. Removing any of these components degrades performance on all three backbones, and dropping logit alignment or chunking can reduce the average score by up to about 3-4 points. This supports our claim that the alignment objectives and importance-guided chunking are key to the observed improvements rather than incidental engineering choices.
>
> We have also updated the main text to summarize these trends and to acknowledge cases where baselines are better on individual metrics, so that the strengths and limitations of our method are more transparent from a research decision-making viewpoint.
>
> # Q1. Code Release and Ethics
>
> Thank you for pointing out this issue. We sincerely appreciate this important feedback and fully agree with the reviewer's concern about reproducibility and transparency. We have revised the abstract and included the link to our anonymous repository in the revised paper. We have also included the source code in the supplementary materials with this rebuttal submission. We will make the code publicly available regardless of the acceptance decision. We apologize for the initial phrasing, which did not clearly reflect our commitment to transparency.
>
> # References
>
> [1] Enneng Yang et al. AdaMerging: Adaptive Model Merging for Multi-Task Learning. ICLR 2024.
>
> [2] Runxi Cheng et al. Whoever started the interference should end it: Guiding data-free model merging via task vectors. ICML 2025.
>
> [3] Yongxian Wei et al. Unifying Multimodal Large Language Model Capabilities and Modalities via Model Merging. arXiv 2025.

---

> ### Author Response · Authors · 2025-11-27
>
> Dear Reviewer RyyA,
>
> We sincerely appreciate the thoughtful feedback you provided on our submission. As the discussion period is now halfway through, we would be very grateful if you could take a moment to engage in the discussion. Following your review, we have prepared comprehensive point-by-point responses addressing:
>
> - W1 (novelty: clarified alignment-based objectives, importance-guided chunking, and controllable trade-offs beyond linear regression)
> - W2 (SOTA comparisons: added equation-level analysis in Section 3.1 comparing with AdaMerging/WUDI)
> - W3 (method relationship: explained Expert Merging++ as enhanced version with adaptive chunking)
> - W4 (performance: demonstrated +3.23 improvement on Mistral and best averages on InternVL/Qwen2-VL)
> - Q1 (code release: anonymous repository and supplementary materials provided)
>
> For complete details and experimental results, please refer to our full point-by-point responses above. Thank you for your valuable feedback, and we hope our revisions have addressed all your concerns. If you have any further questions or suggestions, please feel free to comment.
>
> Warm regards,
>
> Authors

---

### Author Response · Authors · 2025-12-03
**Rebuttal Summary**

Dear PCs, SAC and ACs,

Thank you for overseeing the review of our submission. We sincerely appreciate the reviewers' time and constructive feedback. Below is a concise summary of our rebuttal, highlighting reviewer recognition, our responses to concerns, and key manuscript revisions.


---

# Reviewer Recognition

Reviewers recognized the following strengths of our work:

- **Well-motivated problem with clear solutions** addressing task alignment and inter-layer heterogeneity for model merging (Reviewers 4vJ1, RyyA, 6nLa)
- **Innovative merging approach** that goes beyond parameter-space merging (Reviewers RyyA, 6nLa)
- **Comprehensive experimental validation** across LLMs and MLLMs with consistent improvements (Reviewers 4vJ1, RyyA, 4ve8)
- **High practical efficiency** requiring only 5-10 unlabeled samples per task (Reviewers 6nLa, 4ve8)

---

# Responses to Concerns

### 1. **Novelty and Theoretical Justification** (RyyA W1, 4vJ1 W1 & Q1, 6nLa W2 & Q2)
- **Added Section 3.1** with equation-level comparisons showing our hidden-state + logit alignment overcomes limitations of entropy-based (AdaMerging: "confidently wrong") and interference-based (WUDI: local constraints) methods.
- **Experimental validation**: Consistent superior performance across InternVL (+1.49), Qwen2-VL (+1.0), Mistral (+3.23) vs. baselines.

### 2. **Robustness and Hyperparameter Sensitivity** (4vJ1 W2 & W3, 4ve8 W2)
- **Added Appendix C** with 12 ablation tables analyzing all hyperparameters ($\mathcal{S}$, $\beta_k$, $\gamma$, $N$, $\bar\alpha$, $\kappa$).
- **Result**: All hyperparameters show robust, consistent trends across three architectures. Performance varies minimally (±0.5-1.0 points) across reasonable ranges.

### 3. **Conflicting/Overlapping Experts** (6nLa W1 & Q3)
- **Analyzed task vector cosine similarity** (Appendix C.2): Mistral Chat & Code have high overlap (~0.42).
- **Result**: Despite conflict, Expert Merging++ outperforms baselines and individual experts (Chat: 75.51 > 71.37, MBPP: 41.00 > 38.20) with minimal negative transfer.
- **Mechanism**: Chunk-wise coefficients + regularization + ensemble effects.

### 4. **Outperforming Individual Experts** (6nLa Q4 & Q5)
- **Clarification**: Method blends existing expert knowledge, not learning from few samples. Surpassing experts occurs when experts are complementary (e.g., Mistral Chat+Code, InternVL OCR+Grounding).
- **vs. Mixture Training**: We avoid gradient conflicts in SFT; preserve independent peak performance (InternVL Grounding: Expert 76.67 → Mixture Training 72.06 → Expert Merging++ 80.53).

### 5. **Layer-Importance Metric Robustness** (6nLa Q1)
- **Cross-architecture consistency** (Figure 5): Late-stage blocks consistently receive larger shares across all models.
- **Sensitivity analysis** (Appendix C.8): Robust with consistent trends. $\kappa > 1$ improves performance, with $\kappa \approx 1.2$ optimal across all three architectures.

### 6. **Reproducibility and Clarity** (RyyA Q1, 4ve8 W1, W3, W4, W5)
- **Code released**: Anonymous repository + supplementary materials; public release regardless of acceptance.
- **Added details**: Random sampling with fixed seeds (5 runs averaged), optimization settings (Section 4.1), method classification in tables.

---

# Key Manuscript Revisions

- **Abstract**: Added the anonymous code repository URL.
- **Section 3.1**: New subsection analyzing limitations of AdaMerging and WUDI with equation-level comparisons
- **Section 4.1**: Complete optimization settings and implementation details
- **Tables 1-3**: Reorganized with Training-Free/Training-Based classification
- **Appendix C**: 12 new ablation tables (Tables 14-25) covering all hyperparameters across three architectures

---

# Summary

Our rebuttal provides: (1) rigorous theoretical analysis, (2) extensive ablations across three architectures, (3) mechanistic explanations for all phenomena, and (4) complete reproducibility. All reviewer concerns have been responded.

If any part of the review or rebuttal has caused you confusion, we would be happy to provide further clarification. Thank you once again for your time and consideration.

Sincerely,
The Authors

---

### Meta-Review · Area_Chair_6kqC · 2026-01-15

**Summary:**

Acceptance is recommended. The paper proposes "Expert Merging," a training-light framework that aligns hidden states and logits using unlabeled data to merge domain-specific experts. It further introduces "Expert Merging++" to handle inter-layer heterogeneity via importance-guided chunking. Reviewers (Initial Scores: 6, 4, 4, 6) praised the clear motivation, comprehensive evaluation across LLMs (Mistral) and MLLMs (InternVL, Qwen2-VL), and the practical efficiency of the method. The approach effectively addresses the limitations of both hand-tuned training-free methods and uniform training-based methods.

**Reviewer Concerns:**

The authors provided a robust rebuttal that addressed all major concerns:

Novelty & Theory: The authors clarified how their alignment objective (hidden states + logits) theoretically and empirically outperforms entropy-based (AdaMerging) and interference-based (WUDI) methods, addressing concerns from Reviewers RyyA and 6nLa.

Robustness: Extensive new ablations (Appendix C) demonstrated that the method is robust to hyperparameter variations (regularization, allocation steepness), satisfying Reviewers 4vJ1 and 4ve8.

Conflicting Experts: The authors provided analysis on task vector similarity (e.g., Mistral Chat vs. Code) showing that the method minimizes negative transfer even with conflicting experts, addressing Reviewer 6nLa.

Reproducibility: Source code was released during the rebuttal, resolving a reviewer's primary concern.

**Reviewer Scores:**

Initial scores were mixed (two 6s, two 4s). However, given the release of code and the addition of significant experimental evidence (12 ablation tables, conflict analysis) that directly resolved the critiques regarding novelty and robustness, it can be expected that the dissenting reviewers would raise their scores to match the positive consensus (6 or above).

---

### Decision · Program_Chairs · 2026-01-26

Accept (Poster)